# The neural dynamics of positive and negative expectations of pain

Christoph Arne Wittkamp[†], Maren-Isabel Wolf[†], Michael Rose*

Department of Systems Neuroscience, University Medical Center Hamburg Eppendorf, Hamburg, Germany

## eLife Assessment

Wittkamp et al. investigated the spatiotemporal dynamics of expectation of pain using an original fMRI-EEG approach. The methods are solid and the evidence for a substantially different neural representation between the anticipatory and the actual pain period is **convincing**. These **important** findings are discussed within a general framework that encompasses their research questions, hypotheses, and analysis of results. Although the choice of conditions and their influence on the results might accept different interpretations, the manuscript is strong and contributes beneficial insights to the field.

**\*For correspondence:**
rose@uke.de

[†]These authors contributed equally to this work

**Competing interest:** The authors declare that no competing interests exist.

**Abstract** Pain is heavily modulated by expectations. Whereas the integration of expectations with sensory information has been examined in some detail, little is known about how positive and negative expectations are generated and their neural dynamics from generation over anticipation to the integration with sensory information. The present preregistered study employed a novel paradigm to induce positive and negative expectations on a trial-by-trial basis and examined the neural mechanisms using combined EEG-fMRI measurements (n=50). We observed substantially different neural representations between the anticipatory and the actual pain period. In the anticipation phase i.e., before the nociceptive input, the insular cortex, dorsolateral prefrontal cortex (DLPFC), and anterior cingulate cortex (ACC) showed increased activity for directed expectations regardless of their valence. Interestingly, a differentiation between positive and negative expectations within the majority of areas only occurred after the arrival of nociceptive information. FMRI-informed EEG analyses could reliably track the temporal sequence of processing showing an early effect in the DLPFC, followed by the anterior insula and late effects in the ACC. The observed effects indicate the involvement of different expectation-related subprocesses, including the transformation of visual information into a value signal that is maintained and differentiated according to its valence only during stimulus processing.

## Introduction

Nociceptive input can result in highly variable sensations of pain with expectations playing a crucial role in pain modulation (*Atlas and Wager, 2012*). Positive expectations can lead to hypoalgesia (placebo effect), while negative expectations can increase the perceived intensity of pain (nocebo effect; *Kong and Benedetti, 2014*). Although many studies have shown that expectations can influence pain perception, the neuronal processes underlying the generation of expectations prior to the appearance of the pain stimulus are not yet fully understood (*Benedetti, 2014*; *Büchel et al., 2014*; *Koyama et al., 2005*; *Wager and Atlas, 2015*). Several studies have demonstrated that placebo and nocebo effects influence brain activity during pain perception (*Egorova et al., 2015*; *Wager and Atlas, 2015*; *Zunhammer et al., 2021*). This may occur through multiple pathways that are involved

in integrating expectations and sensory information (*Geuter et al., 2017*). Especially the descending pain modulatory system (DPMS; *Geuter et al., 2017*; *Tu et al., 2022*) is associated with placebo and nocebo effects and is thought to consist of areas like the periaqueductal gray (PAG), dorsolateral prefrontal cortex (DLPFC), anterior cingulate cortex (ACC), and amygdala (*Eippert et al., 2009*), as well as frontal areas like the vmPFC (*Geuter et al., 2017*; *Tu et al., 2022*). Furthermore, placebo and nocebo effects modulate activity in areas classically associated with pain processing like the thalamus and the insula during noxious stimulation (*Atlas and Wager, 2014*; *Wager and Atlas, 2015*; *Wager et al., 2004*; *Zunhammer et al., 2021*).

A central feature of expectations is that they are generated prior to the appearance of the stimulus and should, therefore, be reflected in anticipatory neural activity (*Kong et al., 2007*; *Wager et al., 2004*). The mere expectation of the appearance of a painful stimulus has been shown to activate regions relevant to subsequent pain processing, such as the insula, DLPFC, and thalamus (*Palermo et al., 2015*; *Ploghaus et al., 1999*). Similarly, there is evidence that expecting reduced pain (e.g. via a placebo) modulates activity in parts of the DPMS and the insula already during pain anticipation (*Wager et al., 2011*). This includes the DLPFC (*Amanzio et al., 2013*; *Geuter et al., 2013*; *Wager et al., 2004*; *Watson et al., 2009*) and the ACC (*Amanzio et al., 2013*; *Geuter et al., 2013*) and takes place prior to the widespread modulation of neural activity in the DPMS during pain processing (*Amanzio et al., 2013*; *Atlas and Wager, 2014*). The activation of brain areas prior to the administration of a painful stimulus aligns with the general framework of placebo effects and nocebo effects proposed by *Büchel et al., 2014*. This framework posits that placebo hypoalgesia and nocebo hyperalgesia can be attributed to predictive coding, suggesting that perception is the result of a constant matching of incoming sensory data with the top-down predictions of an internal or generative model (*Büchel et al., 2014*). These top-down predictions should be reflected in the expectation generation happening before the stimulus. However, the neural mechanisms of this process remain unclear, with little information about where and how the expectations relevant to these top-down predictions are generated. Furthermore, it is uncertain whether the anticipatory modulation reported in the literature reflects only a pre-activation of later relevant networks or indicates functionally distinct processes in expectation generation. Comparing neural processing during both the anticipatory and pain periods prior to the stimulus is crucial for better dissociating the formation and maintenance of pain-related expectations from their integration with nociceptive input (*Wager et al., 2004*).

Another important factor when examining the representation of expectations is their valence, i.e., whether they are positive (as in placebo effects) or negative (as in nocebo effects). It remains elusive whether positive and negative expectations share a common neural basis or depend on different networks (*Freeman et al., 2015*). On the basis of behavioral differences between positive and negative expectations, such as their varying correlations with the amount of prior experience and differences in learning (*Colloca et al., 2010*; *Colloca et al., 2008*), it seems reasonable to assume at least partially dissimilar neural representations of positive and negative expectations. Especially during pain perception, there is evidence of distinct neural processes for positive and negative expectations, as well as valence-dependent modulation of similar systems. More specifically, some findings suggest differential modulation of activity in key areas of the DPMS and reward system by the valence of expectations (*Benedetti et al., 2020*; *Crawford et al., 2021*; *Koyama et al., 2005*; *Scott et al., 2008*). Alternatively, some studies have reported the absence of shared brain activations during the perception of pain (*Bingel et al., 2011*; *Freeman et al., 2015*; *Fu et al., 2021*; *Shi et al., 2021*; *Shih et al., 2019*). Similar findings have been reported during pain anticipation, as some areas have been reported to be specifically activated for either placebo or nocebo (*Fu et al., 2021*; *Rossettini et al., 2023*) or show opposing valence-dependent responses (*Amanzio et al., 2013*; *Kong et al., 2008*; *Palermo et al., 2015*). However, there is conflicting evidence of shared anticipatory activation for both placebo and nocebo in some areas of the DPMS like the DLPFC and ACC (*Amanzio et al., 2013*; *Amanzio and Palermo, 2019*; *Colloca and Benedetti, 2007*; *Frisaldi et al., 2015*; *Manaï et al., 2019*; *Palermo et al., 2015*; *Rossettini et al., 2023*; *Schmid et al., 2015*). This illustrates an ongoing debate about whether there is a common neural basis for positive and negative expectations instead of entirely separated representations (*Freeman et al., 2015*; *Fu et al., 2021*). Taken together, the neuronal representations of positive and negative expectations may consist of shared valence-neutral processes (common effects) and different or differentially modulated valence-dependent processes (distinct effects), and this relationship may change over time from the formation of expectations until

their integration with the sensory information. To adequately examine common and distinct neuronal representations, it is imperative to compare positive and negative expectations against each other, as well as to an appropriate control condition without any directed expectation, meaning that perception is not biased in any direction in this condition. Hence, in this study, we implemented a within-subjects design in which participants were subjected to positive, negative, or neutral expectations on a trial-by-trial basis, enabling an exploration of common and distinct processes.

Even during the anticipation phase, it is reasonable to presume the involvement of distinct processes unfolding in a temporal sequence. For instance, the visual cue must first be encoded and transformed into an expectation signal that can be interpreted by a 'pain system'. A potential candidate for this integration process is the insula, which is recognized for its function as a multimodal network hub (*Adler-Neal et al., 2019*; *Dionisio et al., 2019*; *Lu et al., 2016*). The multimodal role of the insula is further reflected in its role in fear conditioning, with the insula being associated with threat (*Fullana et al., 2016*), while other areas that are important for placebo modulations like the vmPFC are closely connected to the default mode network, potentially reflecting a safety signal (*Fullana et al., 2016*). Within the DPMS, prefrontal areas have been suggested to provide predictions for downstream pain-sensitive systems, implying an early role within this system (*Geuter et al., 2017*; *Koban et al., 2017*; *Wager and Atlas, 2015*). Subsequently, it appears plausible that other areas of the DPMS would need to be 'informed' and activated in close temporal proximity to the pain stimulus in order to exert their influence on pain perception (*Amanzio et al., 2013*). The assessment of the temporal profile of expectancy generation is beyond the possibility of fMRI. We therefore combined fMRI with simultaneous EEG measurements, allowing us to temporally localize neural activity by utilizing the temporal and spatial advantages of both techniques at the single-trial level.

In this study, we investigated the neural basis of the common and distinct processes underlying positive and negative expectations, and the formation and integration of expectations into pain perception, using a novel paradigm that allowed the manipulation of expectations on a trial-by-trial basis, while EEG and fMRI measures were recorded. We presented cues to induce expectations (positive, negative, or neutral expectations) followed by an anticipatory period in which different expectations emerged. This allowed us to examine the distinct and common effects of placebo and nocebo in the anticipation and pain phase by comparing the different expectation conditions. We focused on the evaluation of the different neuronal processes that contribute to the generation of directed expectations (i.e. positive and negative) in the anticipatory period and the effects during pain perception, using combined EEG and fMRI.

Based on the literature and the assumed theoretical approach of predictive coding, proposing an expectation formation before a stimulus, we expected that representations of pain-related expectations undergo dynamic changes during the anticipation phase and pain phase, reflecting different processes during these phases such as expectation formation, expectation integration, and pain modulation. These processes could involve either separate networks during the anticipation and pain phase or they could take place in the same networks, with the anticipatory activity having preparatory qualities for the later perception. Furthermore, different patterns of activity for positive and negative expectations could arise. If the valences do not differ in their activation patterns either during pain anticipation or pain processing, this would mark similar processes for both positive and negative expectations. On the other hand, different effects would either indicate distinct processes or a different modulation of the same process. We would mainly expect a distinct nature of the valences, but that similar areas would be engaged throughout anticipation and pain processing. Therefore, we hypothesized to see neural representations of expectations in similar areas during the anticipation and pain phase, albeit that those positive and negative valences would be differentially represented, marking dissociable dynamics of positive and negative expectations.

## Results

In total, we investigated 50 participants (32 female) in a combined EEG-fMRI paradigm. In short, participants were told that they would be given real-time visual feedback on their current pain sensitivity based on their EEG activity using a Brain-Computer Interface (BCI). The feedback indicated one of three different brain states: either a state of high pain sensitivity (red cue/nocebo condition/negative expectation), low pain sensitivity (green cue/placebo condition/positive expectation), or that the BCI algorithm would not make any prediction (yellow cue/control condition/neutral expectation). In fact, the visual

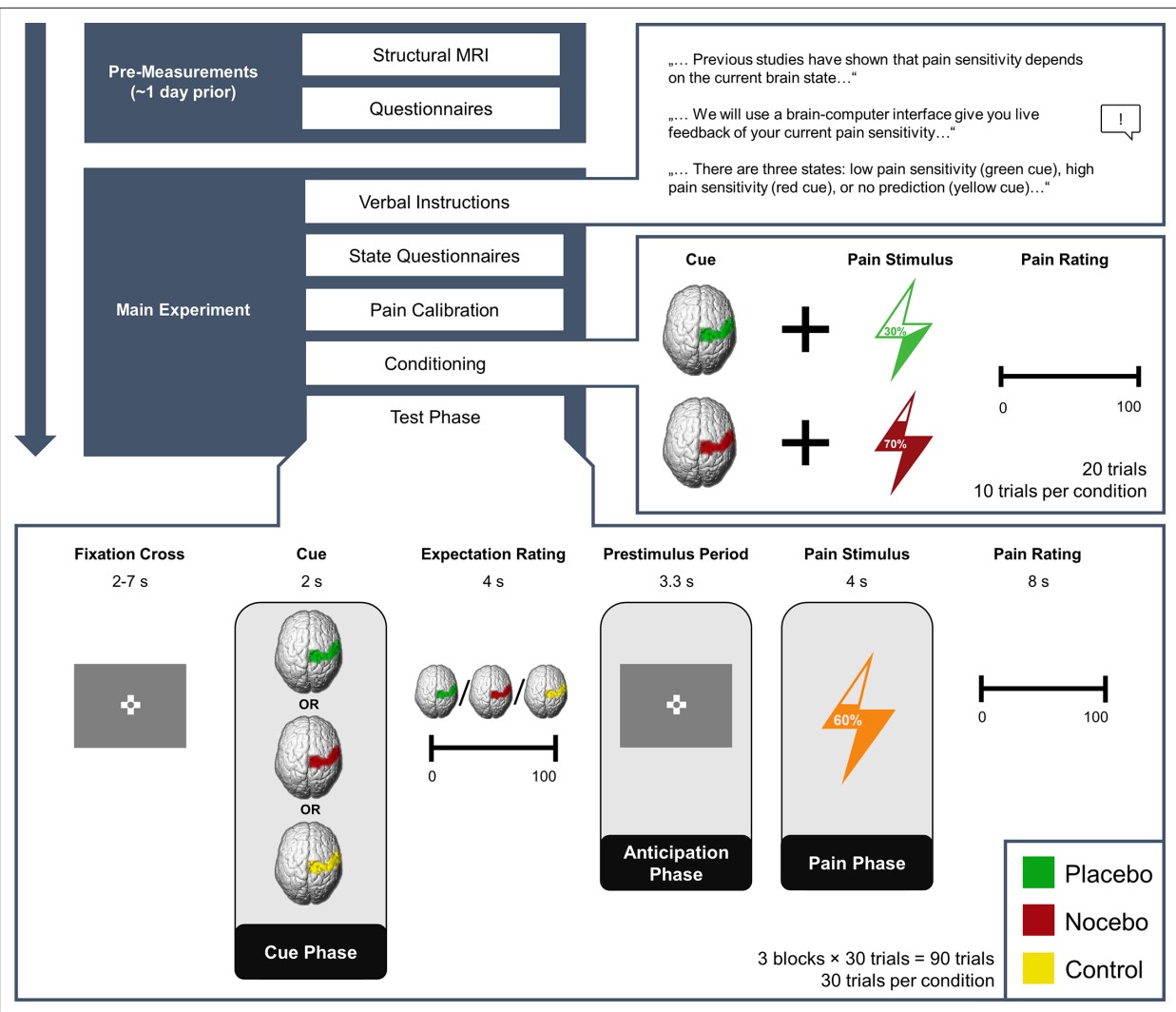

**Figure 1.** Experimental Procedure. Structure of the experiment including pre-measurements and the main experiment. Expectations were generated using a sham brain-computer interface (BCI), i.e., participants were told that they would receive real-time feedback regarding their pain sensitivity (verbal instructions) and experienced the validity of this feedback (conditioning). In the conditioning phase, green cues were paired with lower pain intensities compared to red cues unbeknownst to the participants. In the test phase, the stimulation temperature was always the same, regardless of the cue. The presentation of the condition cue varied from trial to trial.

cues were not related to any brain activity but were only used to produce the corresponding expectations. To reinforce expectations, we also performed a learning (i.e. conditioning) phase. Here, red cues were paired with higher pain intensity (VAS level 70), while green cues were paired with lower pain intensity (VAS level 30). In the ensuing test phase (see *Figure 1*), temperatures were kept constant (VAS level 60). Participants were informed that they would receive different pain stimuli of medium intensity and were unaware that the stimulation temperature was always exactly the same. In each trial, participants were given a BCI-based feedback supposedly related to their current brain state (cue phase) and subsequently rated their pain expectation for the next stimulus (expectation rating). After a fixed anticipation phase, they were presented with a brief heat pain stimulus with a constant target temperature irrespective of condition (pain phase), and lastly had to rate how intensely they perceived the stimulus (pain rating). Apart from EEG and fMRI we also continuously recorded electrodermal activity.

## Successful induction of placebo and nocebo effects in behavioral ratings and skin conductance responses

Our data showed successfully induced expectations in line with the cued sham brain states as evidenced by a significant main effect of condition in a repeated-measures ANOVA for mean expectation ratings

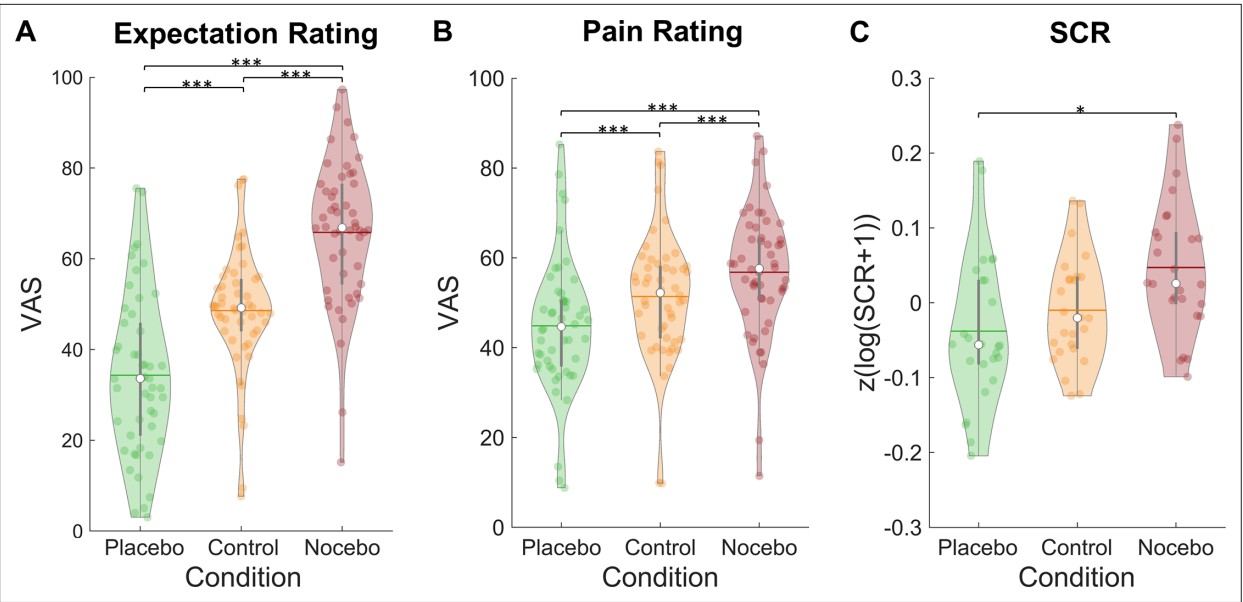

**Figure 2.** Expectation ratings, pain ratings, and skin conductance response by condition. Mean expectation (**A**) and pain ratings (**B**) on a visual analogue scale separately for each condition (n = 50). (**C**) Mean skin conductance responses in the three conditions (n = 26). White dots = mean, horizontal lines = median, thick gray vertical lines = upper and lower quartile, coloured dots = pain ratings of individual participants per condition.*p<0.05. **p<0.01. ***p<0.001.

($F_{(2,98)}$ = 86.51, p<0.001, $\eta_P^2$=0.64; see *Figure 2A*). Expectation ratings were higher in the nocebo (M=65.80, SD = 15.80) compared to the control condition (M=48.58, SD = 13.79, p<0.001), which in turn were higher than in the placebo condition (M=34.33, SD=17.71, p<0.001). Similarly, mean pain ratings were affected by our manipulation (rmANOVA: $F_{(2,98)}$ = 63.00, p<0.001, $\eta_P^2$=0.56; see *Figure 2B*). Post-hoc Tukey tests revealed higher pain ratings in the nocebo (M=56.80, SD=14.21) compared to the control condition (M=51.40, SD=14.31, p<0.001), which in turn led to higher pain ratings than the placebo condition (M=44.88, SD=15.06, p<0.001). Moreover, placebo (control - placebo) and nocebo (nocebo - control) effects were significantly correlated across subjects for both expectation (r=0.64, p<0.001) and pain ratings (r=0.30, p=0.033), indicating that subjects who experienced stronger placebo effects also experienced larger nocebo effects.

To assess whether ratings within the three conditions were stable or varied over time, we compared the relative variability index (*Mestdagh et al., 2018*), a measure that quantifies intra-subject variation over multiple ratings, between the three conditions and over the three measurement blocks. We observed differences in relative variance indices between conditions for both expectation (F(2,96) = 8.14, p<0.001) and pain ratings (F(2,96) = 3.41, p=0.037). For both measures, post-hoc tests revealed that there was significantly more variance in the placebo compared to the control condition (both $p_{holm}$<0.05), but no difference between control and nocebo. Variance in expectation ratings decreased from the first block compared to the other two blocks (F(1.35,64.64)=5.69, p=0.012; both $p_{holm}$<0.05), which was not the case for pain ratings. There was no interaction effect of block and condition for neither expectation (F(2.65,127.06)=0.40, p=0.728) nor pain ratings (F(4,192) = 0.48, p=0.748), which implies that expectations were similarly dynamically updated in all conditions over the course of the experiment.

The expectation manipulation not only affected behavioral ratings but also the skin conductance responses (SCRs) to the pain stimuli (rmANOVA: F(2,50) = 4.33, p=0.018, $\eta_P^2$=0.15; see *Figure 2C*). A post-hoc Tukey test showed larger SCRs in the nocebo (M=0.05, SD=0.09) compared to the placebo condition (M=–0.04, SD=0.10, p=0.049). SCRs in the control condition (M=–0.01, SD=0.07) did not significantly differ from neither the nocebo (p=0.072) nor placebo condition (p=0.607).

## Successful induction of expectation effects in fMRI pattern

Induction of expectation effects was also tested in functional imaging data. For all fMRI analyses, a finite impulse response (FIR) model was used to characterize BOLD fluctuations over time from cue

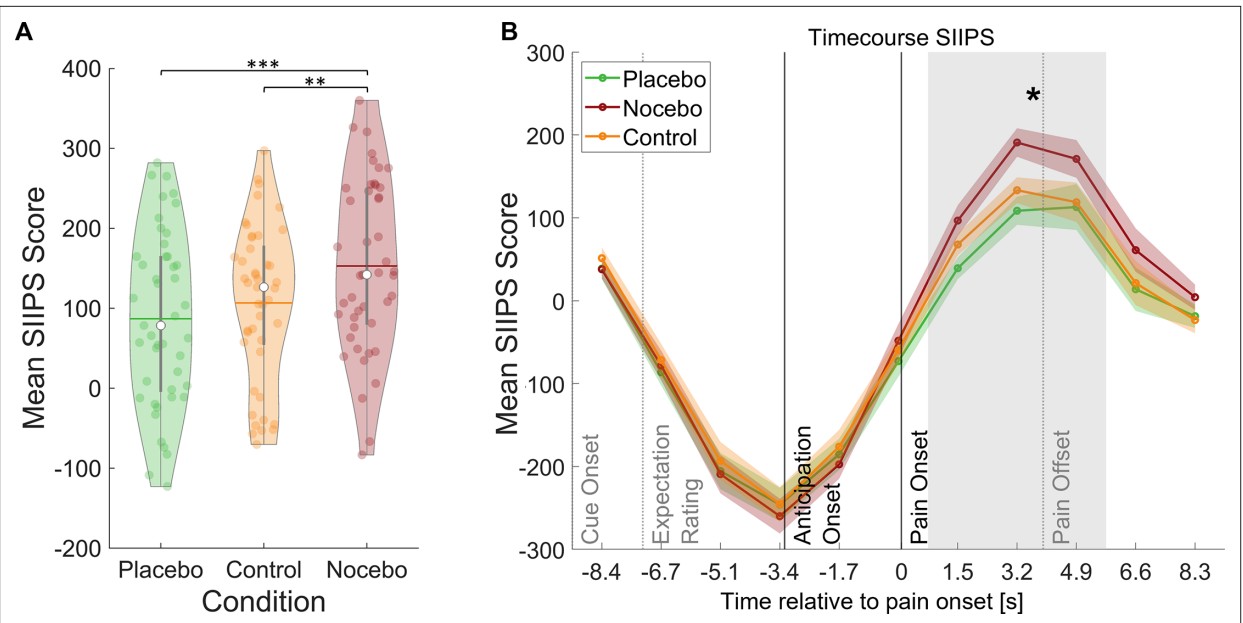

**Figure 3.** Stimulus intensity independent pain signature (SIIPS) scores by condition. (**A**) Mean SIIPS score per condition for all time-points during pain perception. White dots = mean, horizontal lines = median, thick gray vertical lines = upper and lower quartile, coloured dots = pain ratings of individual participants per condition. *p<0.05. **p<0.01. ***p<0.001. (**B**) Mean SIIPS score per condition plotted over the duration of the whole trial. The mean SIIPS scores shown in **A** were extracted from the gray-marked period. n = 45.

onset to the pain rating (see Methods for details). The stimulus intensity independent pain signature (SIIPS) has been introduced as a marker for subjective pain perception going beyond intensity differences as it has been reported to be affected by psychological factors such as expectations (*Botvinik-Nezer et al., 2023*; *Woo et al., 2017*). To further validate our experimental design, we estimated the SIIPS score for each condition during the pain phase as a marker for differences in pain perception between the three conditions (see Methods). A rmANOVA revealed significant differences between the three conditions within the pain period ($F(2,88) = 11.59$, p<0.001, $\eta_P^2=0.21$), with Bonferroni-corrected paired *t*-tests showing significant differences between the placebo and nocebo condition ($t_{(44)} = 4.79$, p<0.001) and between the nocebo condition and the control condition ($t_{(44)} = 3.36$, p=0.002) but not between the control and the placebo condition ($t_{(44)} = 1.35$, p=0.184, see *Figure 3*). We therefore conclude that the manipulation of expectations led to significant perceptual differences. Contrastingly, the SIIPS signature failed to discern between conditions during the anticipation period, suggesting fundamentally distinct processes in the two phases ($F(2,88) = 0.79$, p=0.455, $\eta_P^2=0.02$).

## Neuronal representation of expectations over time

In our main analysis, we found a clear dissociation between the anticipation and pain phases with a predominantly common representation of positive and negative expectations during the anticipation phase and a later shift towards distinct effects during the pain phase (see *Figure 4A*). In order to investigate how the representations of directed expectations changed over time from the anticipation to the pain phase, we identified common (i.e. positive and negative vs. control; constrained to areas with no statistical difference between positive and negative) and distinct (positive vs. negative) effects of directed expectations in each phase (see *Supplementary file 1a-d* for all comparisons). During the anticipation phase, common effects of directed expectations were found in several important areas of the DPMS, e.g., in the bilateral DLPFC, bilateral ACC, and right vmPFC, indicating that directed expectations were represented in a rather general and nonspecific way during this period. With the stimulus onset, activity in these areas showed differential activation between positive and negative expectations. Further differential activity was observed in e.g., the left insula, amygdala, thalamus, and hippocampus during the pain phase.

Crucially, the bilateral DLPFC, right vmPFC, left anterior insula, and thalamus were engaged during both the anticipation and pain phases (see *Figure 4B*). In all these areas, directed expectations shifted

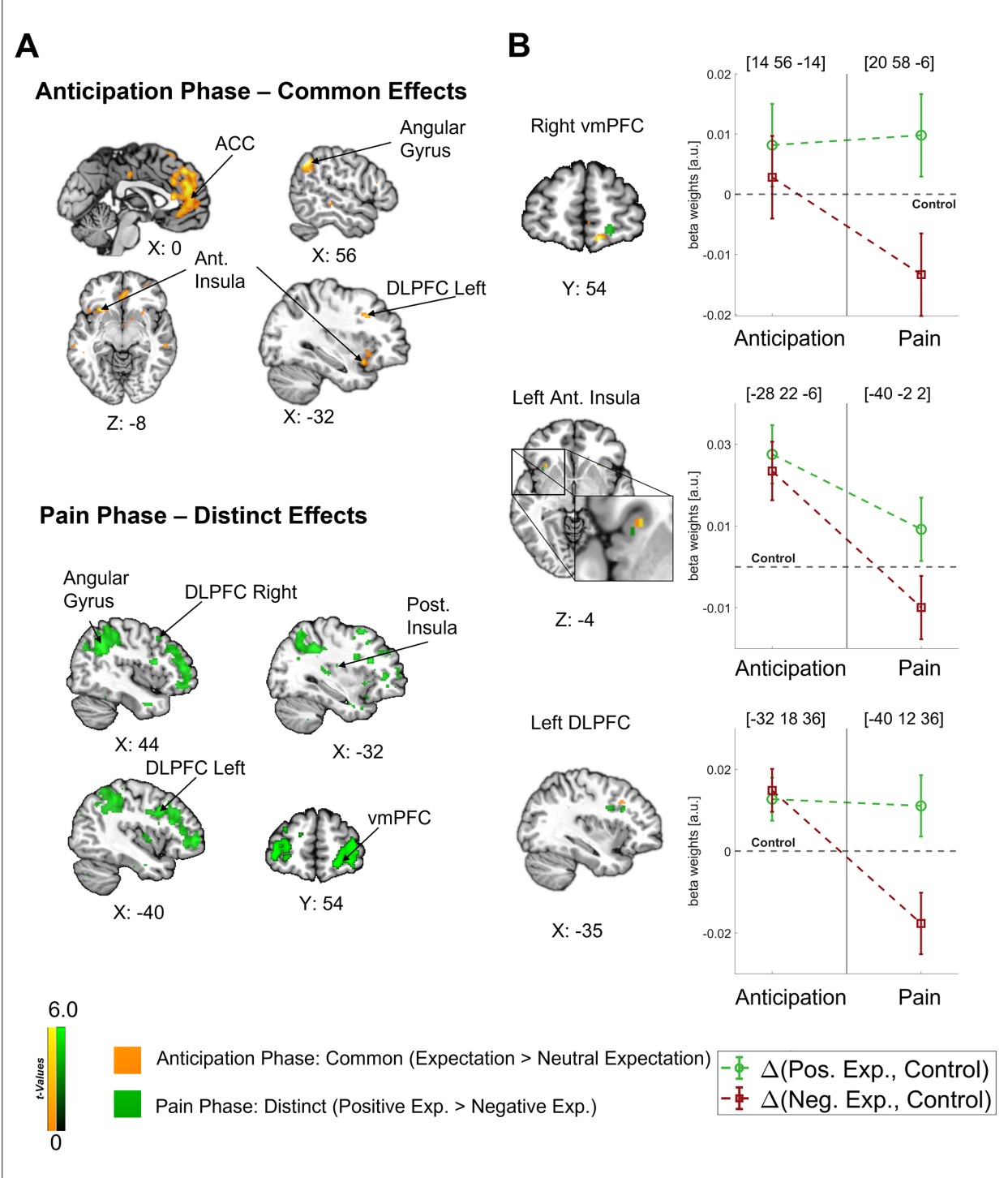

**Figure 4.** Differentiation of effects during the anticipation and pain phase. (**A**) Top: Common effects during pain anticipation (expectation > neutral expectation) at p<0.001 (uncorrected for display purposes) show widespread higher activity for both positive and negative expectations compared to the control condition. **Bottom:** Distinct effects (positive > negative) during pain perception are shown, indicating broadly higher activity for positive compared to negative expectations. (**B**) Left: For selected areas, the overlap between common effects of expectations during the anticipation phase (yellow) and distinct effects of positive and negative expectations during the pain phase (green) in the respective area is shown. Right: The corresponding activation levels of positive and negative expectations (i.e. beta weights from the finite impulse response (FIR) model) baselined by the control condition are plotted for each phase at the respective peaks (peak coordinates in parentheses). The visualization highlights the differentiation of effects following the onset of pain. n = 45.

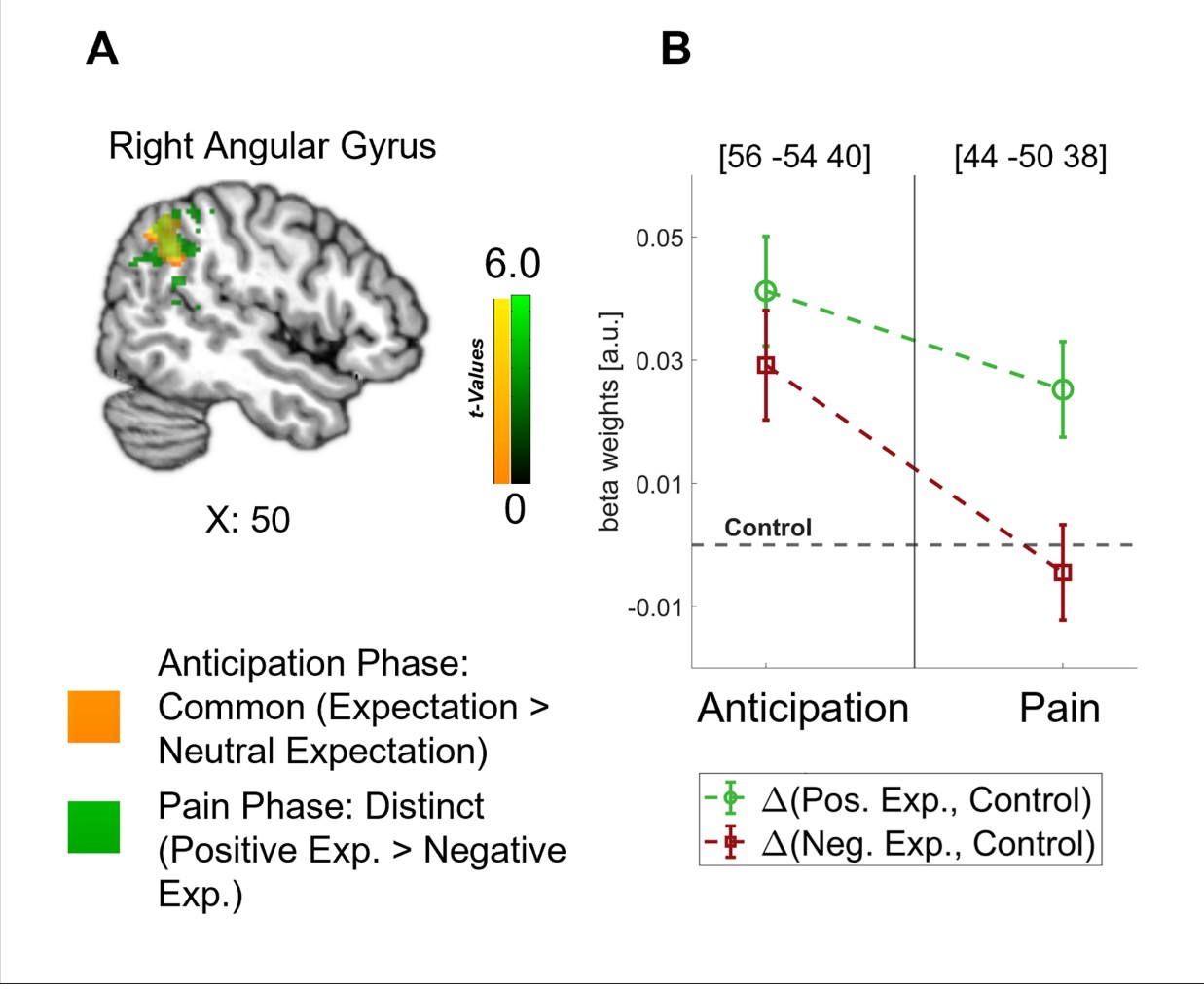

**Figure 5.** Representation of expectations in the angular gyrus. (**A**) Overlap between common effects of expectations during the anticipation phase (expectation > neutral expectation; yellow) and distinct effects of positive and negative expectations during the pain phase (positive > negative; green) is shown for the angular gyrus at p<0.001 (uncorrected for display purposes). (**B**) The corresponding activation levels of positive and negative expectations (i.e. beta weights from the finite impulse response (FIR) model) baselined by the control condition are plotted for each phase at the respective peaks (peak coordinates in parentheses). n = 45.

from a common (positive and negative > control) towards a distinct representation (positive > negative: bilateral DLPFC, right vmPFC, and left anterior insula; negative > positive: thalamus) over time. In addition to these areas that are frequently related to expectation effects, the right angular gyrus was also engaged throughout the time course, similarly initially showing a common representation of positive and negative expectations during the anticipation phase and differentiating only during pain perception (p<0.05 whole-brain FWE-corrected; see *Figure 5*).

The differences between the anticipation and the pain phase demonstrate that specific expectations were mediated by different processes during these phases and arise from a dynamic interplay of brain regions such as the DLPFC, vmPFC, anterior insula, and thalamus over time.

### Timing of effects during the anticipation phase

To obtain detailed information on the temporal characteristics of the expectation effects during pain anticipation, we performed fMRI-informed EEG analyses. Specifically, we were interested in the temporal sequence of the areas involved. Single-trial estimates of fMRI activity during the anticipation phase were correlated with time-frequency decomposed EEG measures for each participant and then statistically tested at the group level. This analysis was conducted separately for the identified regions of interest that represented directed expectations during both the anticipation and pain phase (left

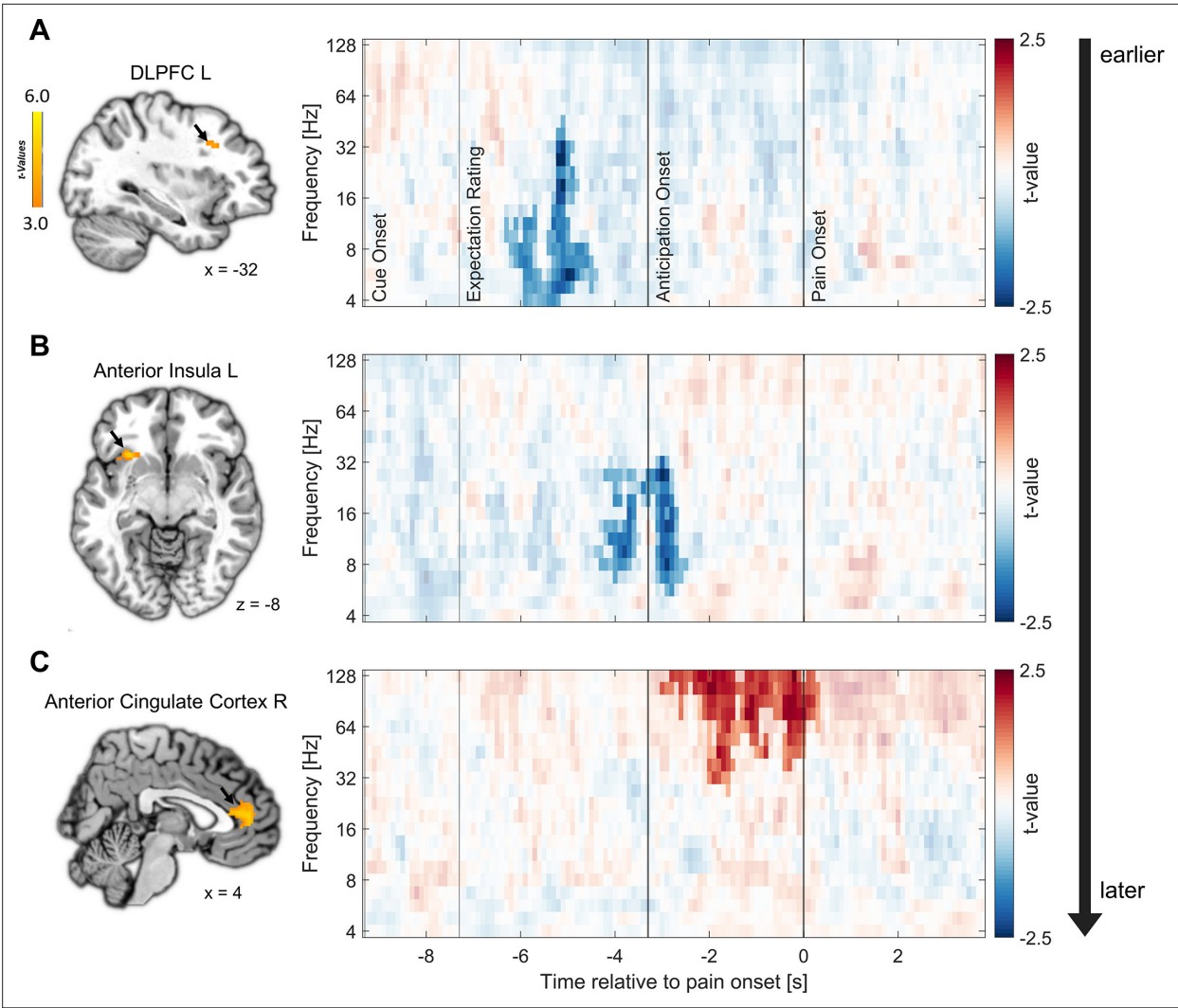

**Figure 6.** Relation of fMRI activity with EEG oscillatory power. Correlation of single-trial hemodynamic responses with time-frequency resolved EEG activity in the left dorsolateral prefrontal cortex (DLPFC) (**A**), left anterior insula (**B**), and right anterior cingulate cortex (ACC) (**C**) during the anticipation phase, ordered by the timing of observed correlations as indicated by the arrow on the right. Single-trial beta weights were extracted from spherical ROIs (10 mm radius) centered around the peak voxels based on the comparison of beta weights from the finite impulse response (FIR) model between expectation and neutral expectation during the anticipation phase, as shown on the left (p<0.001 uncorrected for display purposes). On the right, the cluster-corrected correlation of oscillatory power with fMRI activity averaged over all cluster electrodes is depicted. Non-significant time-frequency points are masked (n = 41).

anterior insula, right vmPFC, bilateral DLPFC, and thalamus). We further included the bilateral ACC, as we were interested in all areas that reflected directed expectations during the anticipation phase.

Clear temporal differences between areas were observed. The earliest correlation of fMRI activity with EEG oscillatory power was found in the left DLPFC already during the expectation rating at theta to low gamma frequencies in a negative direction (−6.3 until −4.4 s, 4–45.28 Hz; p=0.007; see *Figure 6A*; additional information for all regions can be found in *Supplementary file 1e*). Next, anticipatory activity in the left anterior insula was associated with decreased EEG oscillatory power during the late expectation rating and anticipation phase (−4.4 to −2.5 s) spanning from theta to low gamma frequencies (5.67–32 Hz; p<0.001; see *Figure 6B*). Lastly, we found a significant positive association of EEG activity with fMRI activity in the right ACC during the anticipation and early pain phase in the gamma frequency range (−3 to 0.3 s, 26.89–128 Hz; p=0.003; see *Figure 6C*). As expected due to the close spatial proximity, activity in the left ACC was similarly positively correlated with EEG activity in the anticipation and pain phase (cluster 1: −2.3 to −1.3 s, 32–128 Hz; p=0.019; cluster 2: −0.9 to 0.3 s,

32–128 Hz; p=0.021). We did not observe significant correlations of EEG power with fMRI activity in the other ROIs.

In summary, the analyses indicated that areas exhibiting similar fMRI effects for positive and negative expectations during the anticipation phase are linked to distinct temporal and oscillatory patterns. Notably, the left DLPFC and the left anterior insula displayed an early negative correlation between anticipatory activity and EEG power, primarily in the lower frequency range. Conversely, the subsequent effects in the bilateral ACC were associated with an increase in gamma oscillations and were observed at later time points during the anticipation phase indicating different processes.

## Discussion

The manipulation of positive, negative, or neutral expectations on a trial-by-trial basis allowed for a detailed analysis of their neural representations during the generation and integration of expectations with the nociceptive input. Our results revealed fundamental differences between the anticipation and pain phases, indicating the involvement of divergent processes. During the anticipation phase, valence-neutral representations were observed in areas of the DPMS and the anterior insula. After the onset of the nociceptive stimulus, these areas showed differentiated representations indicating separate processes for the formation of expectations and their integration within stimulus processing. The excellent temporal resolution of our fMRI-informed EEG measures further revealed a temporal sequence of expectation processing during the anticipation phase, with an early effect within the DLPFC, followed by activation in the anterior insula, and late effects within the ACC, in line with the occurrence of different time-sensitive sub-processes of expectation generation in this phase.

Our novel within-subjects paradigm was highly effective, as expressed by 47 out of 50 subjects consistently expecting and perceiving the intensity of pain stimuli in line with the cues that induced positive, negative, or neutral expectations. The trial-by-trial manipulation of expectations did not only affect self-reports, but also objective markers such as skin conductance responses and scores of the SIIPS (*Woo et al., 2017*). The SIIPS is an indicator of neural activity that tracks differences in pain perception that go beyond pain intensity and is classically affected by placebo and nocebo interventions (*Botvinik-Nezer et al., 2023*; *Woo et al., 2017*), thus serving as an excellent manipulation check for our paradigm. In line with our expectations, the SIIPS scores discriminated between positive and negative expectations only during the pain phase and not during the anticipation phase. This is a first hint at substantially different processes during the two phases.

The neural representations of directed expectations differed depending on the valence of the expectation, i.e., if they were positive or negative, and the time period. A shift from common (valence-neutral) to distinct (valence-dependent) effects was evident in areas classically involved in placebo analgesia during pain processing, including the bilateral DLPFC, right vmPFC, left anterior insula, and thalamus. The observation of this pattern in key areas of the DPMS indicates that the activation during the anticipation phase is not just mere pre-activation for later pain modulation, but that preparatory processes that are distinguishable from those that occur during pain perception take place. These anticipatory processes may include expectation generation and maintenance, while the focus shifts towards the integration of these expectations into the sensory stream and the evaluation of the percept during pain processing. Importantly, the spatial resolution of fMRI is limited when it comes to discriminating whether the same pattern of activity is due to identical activation or to activation in different sub-circuits within the same area. Nonetheless, the overlap of areas is an indicator of similar processes involved in a more general preparation process. The observed differentiation during pain is in agreement with the few studies that looked into the neuronal representations of positive and negative expectations and mainly found differential effects, e.g., opposite responses or differentially modulated areas during pain (*Benedetti et al., 2020*; *Crawford et al., 2021*; *Koyama et al., 2005*; *Scott et al., 2008*).

The network of the DPMS around the DLPFC, ACC, and vmPFC appears to play a pivotal role in expectation processes and in the valence-dependent modulation of pain perception. Following the framework of predictive coding, our results would suggest that the DPMS is the network responsible for integrating ascending signals with descending signals in the pain domain and that this process is similar for positive and negative valences during anticipation of pain but differentiates during pain processing. One important node of the DPMS is the DLPFC. The outstanding role of the DLPFC in placebo effects is well-established and supported by numerous reports of increased DLPFC activity

during pain anticipation (*Watson et al., 2009*) as well as during pain processing (*Zunhammer et al., 2021*). This area has been classically associated with top-down-regulation and expectation modulation (*Geuter et al., 2017*), as when its neuronal excitability is experimentally manipulated, placebo effects can be diminished or enhanced (*Egorova et al., 2015*; *Krummenacher et al., 2010*; *Tu et al., 2021*). The early onset of DLPFC activity implied by the EEG analysis now suggests that the DLPFC plays an important role in the initiation of both positive and negative expectation effects within the DPMS prior to the noxious stimulation (*Frisaldi et al., 2015*; *Geuter et al., 2017*; *Rossettini et al., 2023*; *Wager and Atlas, 2015*; *Wager et al., 2011*). The vmPFC is another important hub within the DPMS which is suggested to integrate the input from the DLPFC and to generate affective meaning in order to maintain expectations (*Geuter et al., 2017*). It further modulates pain perception by directly affecting brainstem systems (*Koban et al., 2017*), e.g., by influencing PAG activity (*Geuter et al., 2017*). The notion that prefrontal areas exert top-down control over other areas of the DPMS is further supported by the earlier timing of DLPFC effects compared to the ACC (*Craggs et al., 2014*; *Watson et al., 2009*). The ACC also has a direct influence on the activity of the PAG, suggesting that the ACC is another crucial area for pain modulation (*Geuter et al., 2017*; *Livrizzi et al., 2022*). The effect of the bilateral ACC at the transition from the anticipation to the pain phase as indicated by our combined EEG-fMRI analysis could be interpreted as a preparatory mechanism for the modulation of incoming sensory information in downstream areas, consistent with its supposed role in the DPMS (*Geuter et al., 2013*; *Geuter et al., 2017*; *Kong et al., 2008*). Using this framework, reports of ACC activations for both positive and negative expectations during pain anticipation could be understood as a pre-activation of the ACC for pain modulation in either direction (*Rossettini et al., 2023*). The link between anticipatory activity in the ACC and EEG oscillatory activity was observed in the high gamma band, which is consistent with findings that demonstrate a connection between increased fMRI BOLD signals and a relative shift from lower to higher frequencies (*Kilner et al., 2005*). Gamma oscillations have been repeatedly reported in the context of pain and expectations and have been interpreted as reflecting feedforward signals of noxious information (e.g. *Ploner et al., 2017*; *Strube et al., 2021*). In combination with our findings, this might imply that high-frequency oscillations may not only signal higher actual or perceived pain intensity during pain processing (*Nickel et al., 2022*; *Ploner et al., 2017*; *Strube et al., 2021*; *Tu et al., 2016*), but might also be instrumental in the transfer of directed expectations from anticipation into pain processing.

Similarly, a shift from valence-neutral towards valence-dependent processing over time was demonstrated in the anterior insula. The insula is a key brain region involved in the network responsible for pain processing (e.g. *Atlas and Wager, 2014*; *Kober et al., 2008*) but is also part of several non-pain-related networks and works as a multimodal network hub (*Horing and Büchel, 2022*). Results of the fMRI-informed EEG analysis indicated early effects in the anticipatory period consistent with an early role of the anterior insula in expectation generation and initiation. Due to the multimodal nature of the insula, it may be speculated that multiple networks interact to integrate relevant information from different domains (e.g. from the visual cue and interoceptive information) into an expectancy signal in the anterior insula. Due to its connections and function within the salience network, this may involve encoding the expected threat level and salience of subsequent pain stimuli, leading to an anticipatory activation prior to the actual perceptual modulation (*Taesler and Rose, 2016*; *Wiech et al., 2010*). During pain perception, the anterior insula may then be more engaged in a network responsible for stimulus processing and evaluating actual salience and prediction errors in this perceptual process (*Horing and Büchel, 2022*), which may imply that the insula is involved in multiple tasks over time.

Similar to the anterior insula, the activation of the angular gyrus during pain processing and pain anticipation could be understood as an indicator for processes related to the formation and integration of directed expectations. The angular gyrus has been connected to expectation-related pain modulation only a few times (e.g. *Atlas and Wager, 2014*; *Tu et al., 2019*) but its involvement is prominent in our results. Based on its presumed function in maintaining recollected multimodal representations (*Jablonowski and Rose, 2022*; *Vilberg and Rugg, 2012*), the angular gyrus may be engaged in transforming sensory information from the visual cues into expectancy signals that can be processed by e.g., the DPMS and salience network.

With our rather unconventional and new paradigm, we were able to manipulate participants expectations on a trial-by-trial level and derive insights into the neural dynamics of positive and negative expectations. While this may give rise to questions regarding the comparability of our study to

previous paradigms and the manner in which our control condition was employed, we would argue that our expectation manipulation falls in line with a manipulation of treatment expectancies, a typical method of expectation manipulation in placebo paradigms (see *Atlas and Wager, 2014*). Accordingly, it is comparable to previous studies on placebo and nocebo effects. In our study, participants were presented with a cue that induced expectations regarding a 'treatment,' although in this case the 'treatment' originated from changes in their own brain activity. This is, in a broader sense, comparable to studies utilizing sham TENS-devices that are supposedly altering peripheral pain transmission (*Skvortsova et al., 2020*).

Moreover, implementing a proper control condition in expectation modulation paradigms is an inherently difficult task as forming expectations about our environment is a natural process. Therefore, we recognize that participants most likely did form expectations of medium pain intensities in the control condition over the course of the experiment. This is in line with previous research on placebo and nocebo effects, in which participants also typically rated control stimuli in between placebo and nocebo conditions (*Bingel et al., 2011*; *Colloca et al., 2010*; *Shih et al., 2019*). However, we would still argue that we can meaningfully compare the placebo and nocebo condition to the control conditions to investigate the neuronal underpinnings of expectation effects. Independently of whether participants build up an expectation of 'medium' intensities in the control condition, which caused them to perceive stimuli in line with this, or if they simply perceived the stimuli as they were (of medium intensity) with limited effects of expectations, the crucial difference to the placebo and nocebo conditions is that there was no alteration of perception due to previous experiences or verbal information and no shift of perception from the actual stimulus intensity towards any direction in the control condition. Thus, we were able to identify the effects of directed expectations by comparing positive and negative expectations to neutral expectations as a baseline. Our analysis of within-condition variability further showed that ratings indeed varied within conditions and that the amount of variation was comparable between nocebo and control. Over time, expectations were dynamically updated in all three conditions, speaking against alternative explanations of the rating differences between conditions such as a regression to the mean of ratings in the control condition.

Based on the present results, understanding the processing of expectations requires an examination of its temporal and spatial dynamics from anticipation to pain processing, while comparing positive and negative valences to each other and to a control condition. We found largely comparable activation for positive and negative expectations during the anticipation phase, including regions outside of those classically observed in pain processing and modulation. Based on the observed temporal profiles, the DLPFC and anterior insula may be related to the top-down initiation and generation of expectations, while the ACC is activated in close proximity to pain onset as a direct link to pain modulation. During the pain phase, the focus shifts from expectation generation and maintenance towards pain modulation in either positive or negative direction, leading to distinct effects of positive and negative expectations in many areas that initially encoded expectations independently of their valence. It is not surprising that expectations are not a static process but involve different time-dependent components, as pain perception was also recently described as a complex process of interactions among multiple brain systems that are reconfigured over time (*Lee et al., 2022*). Expectation generation, integration, and pain perception all appear to be dynamic processes, with both common and distinct routes for positive and negative expectations, depending on the time point of examination.

## Materials and methods

### Key resources table

| Reagent type (species) or resource | Designation | Source or reference | Identifiers | Additional information |
|---|---|---|---|---|
| software, algorithm | Matlab (2021b) | mathworks.com | RRID:SCR_001622 | |
| software, algorithm | SPM 12 (7771) | https://www.fil.ion.ucl.ac.uk/spm/ | RRID:SCR_007037 | |
| software, algorithm | Ledalab (V3.4.9) | http://ledalab.de/ | | |
| software, algorithm | JASP (0.18.3) | https://jasp-stats.org/ | RRID:SCR_015823 | |

**Table 1.** Characteristics of study participants.

| | Mean | SD | Range | Number (%) |
|---|---|---|---|---|
| Gender | | | | |
| Male | | | | 18 (36%) |
| Female | | | | 32 (64%) |
| Age (years) | 25.4 | 3.5 | 18–34 | |
| FOP | | | | |
| Severe Pain | 36.5 | 5.4 | 22–47 | |
| Minor Pain | 18.9 | 4.9 | 10–33 | |
| Medical Pain | 25.8 | 6.4 | 12–42 | |
| STADI | | | | |
| Anxiety | 15.2 | 4.1 | 10–28 | |
| Depression | 17.0 | 3.2 | 11–25 | |
| Global Score | 32.2 | 5.1 | 23–43 | |
| BDI-II Global Score | 6.0 | 3.8 | 0–16 | |

Note. STADI = State-Trait Anxiety Depression Inventory. FOP = Fear of Pain Questionnaire. BDI-II=Beck Depression Inventory-II.

## Participants

In total, 55 volunteers were recruited via an online job platform and participated in our preregistered study (German Clinical Trials Register; ID: DRKS00025872). All participants were right-handed, had normal or corrected-to-normal vision, reported no neurological or psychiatric diseases, pain conditions, current medication, substance abuse, or pregnancy, and were non-smokers. They gave written informed consent and were compensated with 15 Euros per hour of participation. Of these 55 participants, five had to be excluded from all analyses (four due to technical issues leading to the abortion of the measurement, one due to a severe BDI score), leading to a final sample size of n=50 (see *Table 1*). As preregistered, three participants who rated expectation and pain averaged over the entire experiment higher for placebo compared to nocebo and/or stated that they did not believe in the BCI method were excluded from the analysis of neural data, as we reasoned that the analysis of expectation-related neural activity requires a successful induction of expectations. Analyses for fMRI data were performed additionally excluding two participants with bad MRI data (leading to n=45 for fMRI analyses), and for combined EEG-fMRI analyses additionally excluding four participants with excessive artifacts and/or recording equipment malfunction (leading to n=41 for combined EEG-fMRI analyses). The study was approved by the local ethics committee (PV7170).

## Procedure

### Pre-measurements (Collaborative Research Centre recordings)

One day before the actual study, we recorded fMRI data (T1, functional EPI, DW EPI) and asked the participants to complete a comprehensive psychosocial questionnaire battery that will be analyzed by other projects under the structure of the overarching collaborative research center and are beyond the scope of this manuscript. Participants were pseudonymized using ALIIAS (*Englert et al., 2023*).

### Main experiment

The experiment consisted of four phases: a verbal instruction phase, a pain calibration phase, a conditioning phase, and a test phase. The experiment was programmed using Psychtoolbox3 (http://psychtoolbox.org/) for Matlab (Version R2021b; The MathWorks). Rating responses were given by the participants using a Button Box MR. Instructions and ratings were presented on an MR-compatible monitor with a resolution of 3840×2160, placed at one end of the scanner. Participants saw the monitor through a mirror that was placed approximately 12 cm away from the participant's eyes and

had a distance of approximately 151 cm from the monitor. Two researchers and one radiographer guided the participants through the instructions and preparations.

### Verbal instruction phase

After being prepared for the EEG and fMRI recording, participants were verbally informed that their current oscillatory state of the primary somatosensory cortex would be measured in real-time using a BCI (Brain Computer Interface) and that this state would reflect their pain sensitivity. They were further told that the measured brain state would be visualized in the form of visual stimuli consisting of a brain image with the right primary somatosensory cortex highlighted in one of three different colors (green, red, or yellow). A green stimulus represented a state in which their brain would be less susceptible to pain, a red stimulus represented a state in which their brain would be highly susceptible to pain, and a yellow stimulus represented a state in which the algorithm was not able to detect a clear-cut state and would thus make no prediction (e.g. due to high fluctuations in brain activity or intermediate activity levels). With this procedure, a positive expectation was induced by the green cue, and a negative expectation was induced by the red cue. After the verbal instructions, participants were asked to fill out state questionnaires, including the State-Trait Anxiety Depression Inventory (*Laux et al., 2013*) and the Fear of Pain Questionnaire (*McNeil and Rainwater, 1998*).

### Pain calibration phase

Heat stimuli were delivered with a PATHWAY CHEPS (Contact Heat-Evoked Potential Stimulator) thermode (https://www.medoc-web.com/pathway-model-cheps), which has a rapid heating rate of 70 °C/s and a cooling rate of 40 °C/s and can deliver pain stimuli in the range of 30 to 55°C in less than 300 ms. For all phases, the baseline temperature was set to 32 °C, and the rise and fall rates were set to 70 °C/s. The thermode head was attached to a location directly proximal to the volar mid-forearm. Using a stepwise procedure, we determined individual temperatures for each participant corresponding to values of VAS30, VAS60, and VAS70 on a visual analog scale (VAS) from 0 ('no pain') to 100 ('unbearable pain'). Target temperatures were calculated using linear regression.

### Conditioning phase

For the conditioning phase, the location of the thermode head was changed to a location directly distal to the volar mid-forearm to avoid unnecessary sensitization of one location and skin irritations. Participants were instructed that the next phase would serve as the calibration of the BCI algorithm introduced in the verbal instruction phase. They were informed that in this phase only green and red cues would appear because the pain stimulation would only occur once a clear-cut state of their brain has been detected. Each trial began with the presentation of either a red or green visual cue for 2 s, then the painful stimulus was administered for 4 s, and lastly, participants were asked to rate their pain experience for 8 s. Between trials, there was a fully randomized inter-trial interval (ITI) of between 2 and 7 s. During the painful stimulation and ITIs, a fixation cross was presented in the middle of the screen. Perceived pain intensity was again rated on a VAS from 0 ('no pain') to 100 ('unbearable pain'). Unbeknownst to the participants, green cues were always followed by less painful stimuli (VAS30), and red cues were always followed by more painful stimuli (VAS70). They received 10 stimuli of each condition, leading to 20 trials in total. The order of stimuli was pseudo-randomized with the restrictions of no more than two direct repetitions of the same condition and the last two trials of this phase belonged to the less painful condition.

### Test phase

For the test phase, the thermode head was once again attached to the location directly proximal to the volar mid-forearm. Participants were informed that the BCI algorithm has now been calibrated and would be tested in the next phase. They were told that the painful stimulation would occur at random predetermined points in time, so that either highly pain-sensitive (red; nocebo condition) or less pain-sensitive states (green; placebo condition) could be detected and reported back to the participant, or that they would receive feedback that the algorithm was not able to detect a clear-cut state (yellow; control condition). The trial structure was similar to the conditioning phase, with the change that after cue presentation, participants were asked to rate how painful they expected the next stimulus to be

on a VAS ranging from 0 to 100 while the cue was still presented on the screen (4 s). After this expectation rating, a fixation cross was presented for 3.3 s (anticipation phase) before the pain stimulus was administered for 4 s (pain phase). Independently of the cue color, participants always received painful stimuli corresponding to values calibrated to VAS60. Importantly, participants were only informed that they would receive different stimuli of medium intensity and were thus not aware that the stimulation temperature remained constant. There were 30 cues of each condition followed by pain stimulation divided into three blocks, summing up to a total of 90 stimuli. Similarly to the conditioning phase, the ITI was fully randomized between 2 and 7 s. The order of cues was pseudo-randomized with no more than two direct repetitions of the same condition. Before each block, we applied one pain stimulus of VAS60 without a cue to desensitize the new skin area.

### Follow-Up
One week after the main experiment, participants were invited for a follow-up measurement which is beyond the scope of this manuscript. They were asked to fill out questionnaires, including the Beck Depression Inventory-II (*Beck et al., 1996*; *Hautzinger et al., 2006*). Lastly, participants were debriefed and paid.

## Data acquisition
### Electrodermal data
Electrodermal activity was measured with MRI-compatible electrodes on the thenar and hypothenar. Electrodes were connected to Lead108 carbon leads (BIOPAC Systems, Goleta, CA). The signal was amplified with an MP150 analogue amplifier (also BIOPAC Systems) and sampled at 5000 Hz using a CED 1401 analogue-digital converter (Cambridge Electronic Design, Cambridge, UK).

### fMRI data
MRI was performed with a 3T Siemens PRISMA Scanner, and a 64-channel head coil was used. On the day of the pre-measurements, a T1 image with the following parameters was acquired: T1 FLASH 3D: TE 2.98 ms, TR: 2300 ms, matrix flip angle: 9°, FOV 25.6 * 25.6 cm, TA: 7:22 min. Two sequences on the day of the main experiment were acquired: An EPI BOLD sequence and a field map sequence. Participants were prepared with a 64-channel standard BrainCap MR for 3Tesla (2020 Version) and the EPI BOLD sequence had therefore to be adjusted to meet the necessary safety criteria. The following parameters were used: 2 D EPI BOLD: TE: 29.0 ms, TR: 1679.00 ms, FOV: 22.4 * 22.4 cm, flip angle: 70°, s1: 2 mm, TA: 20:17 min, fat saturation, 715 volumes in total; 2 D field map sequence: TR: 594 ms, TE1: 5.51 ms, TE2: 7.79 ms, FOV: 22.4 * 22.4 cm, flip angle: 40°, s1: 2 mm, TA: 1:31 min.

### EEG data
Continuous EEG data was recorded inside the MRI scanner using a custom 64-channel BrainCap-MR for 3 Tesla using BrainVision Recorder (Version 1.10, BrainProducts, Gilching, Germany). The cap contained 64 passive sintered Ag/AgCl electrodes arranged according to the 10/20 System, as well as one ECG electrode. FCz served as the reference and Pz served as the ground electrode. The cap was connected to two Brain Amp MR plus amplifier systems with 32 channels each (BrainProducts, Gilching, Germany), powered by one rechargeable battery unit. Amplifiers and the battery unit were positioned on foam cushions directly behind the head coil inside the scanner. Electrode skin impedance was kept below 10 kΩ. EEG data was recorded with a sampling rate of 5000 Hz and an amplitude resolution of 0.5 µV for EEG channels and 10 µV for the ECG channel. The EEG system was synchronized with the clock of the MRI system using a SyncBox (BrainProducts, Gilching, Germany). The helium pump of the MRI system was switched off during data recording. Data was transmitted from the amplifiers to the recording computer outside of the scanner room via a fiber-optic cable.

## Preprocessing
### Electrodermal data
Preprocessing and analysis of electrodermal data were performed using the Ledalab toolbox for MATLAB (*Benedek and Kaernbach, 2010*). Single-subject data was downsampled to 100 Hz and visually screened. In total, 21 subjects were excluded from the electrodermal analysis (18 due to

physiological non-responsiveness, three due to equipment malfunction). From the remaining 26 subjects, all data segments around pain stimulation were screened for excessive artifacts, resulting in the exclusion of 55 of the 2340 segments (2.35%). Using a deconvolution method implemented in Ledalab, raw electrodermal data were decomposed into continuous phasic (driver) and tonic components. Subsequent analyses were performed on the extracted phasic skin conductance responses (SCRs). The response window for pain was determined by visual inspection of the curve to cover the peak and set between 2 and 7.5 s. SCR segments within the response window were log- and z-transformed within participants. For the log-transform, a constant (minimum of the driver plus 1) was added to the data to shift it to positive values. Lastly, segments were averaged per subject for each of the three conditions.

### fMRI data

Preprocessing of fMRI data was done using the Statistical Parameter Mapping software (SPM 12, Wellcome Department of Imaging Neuroscience, London, UK, https://www.fil.ion.ucl.ac.uk/spm/software/spm12/). The first two volumes of each block were dropped to get full MRI saturation effects. Furthermore, realignment and unwarping, registration to standard space (Montreal Neurological Institute), and spatial smoothing with a 6 mm Gaussian kernel were used on the data.

### EEG data

MR and cardioballistic artifacts were corrected using BrainVision Analyzer 2.2 (BrainProducts, Gilching, Germany) for each block separately. Continuous MR artifacts were corrected with sliding baseline corrected average templates. Data was then downsampled to 500 Hz. Cardioballistic artifact correction was done by semi-automatically detecting a pulse template, marking it in the electrocardiogram channel, and then subtracting it from recordings.

For the remaining preprocessing and analysis, we used the Fieldtrip toolbox for Matlab (*Oostenveld et al., 2011*). Data were cut into trials including all relevant time intervals from 1000 ms before cue onset to the end of pain 15,800 ms after cue onset. The resulting segmented data were low-pass filtered at 150 Hz and high-pass filtered at 0.5 Hz. We adapted a recent preprocessing approach introduced by *Hipp et al., 2011*. The data was split into low- and high-frequency data (34 Hz low-pass filter and 16 Hz high-pass filter, respectively) and processed in parallel. This approach leads to high sensitivity in detecting and removing artifacts from the data as e.g., heartbeats cause more artifacts at lower frequencies and muscle activity affects higher frequencies more strongly. All trials were visually inspected and removed for both subsets when containing large artifacts. Then, both high- and low-frequency data were subjected to an independent component analysis (ICA) using a logistic infomax algorithm. Components reflecting residual cardioballistic and MR artifacts, blinks, eye- and head movement, and muscle activity were identified by visual inspection of the time course, spectrum, and topography of each component and discarded. Both subsets were re-referenced to the average of all channels and the original reference electrode was regained. Lastly, we subjected all data to another full visual scan and shifted the time axis so that the onset of pain stimulation occurred at t=0 s. In total, the visual artifact screening led to the exclusion of 228 of the 3944 recorded trials (5.78%).

### Time-frequency decomposition

Our procedure was again adapted from *Hipp et al., 2011*. Time-frequency decomposition was conducted for 21 logarithmically spaced frequencies ranging from 4 to 128 Hz (0.25-octave increments) in 0.1 s steps using the multi-taper method based on discrete prolate spheroid sequences (DPSS). The high-frequency data were used for the frequency transformation of frequencies above 25 Hz, and the low-frequency data for frequencies below 25 Hz. Temporal and spectral smoothing were adjusted to match 250 ms and 3/4 octave, respectively. This was achieved by fixing the time window to 250 ms and adjusting the number of Slepian tapers for frequencies larger than 16 Hz, while for frequencies up to 16 Hz, a single taper was used, and the time window was adjusted. We extracted single-trial time-frequency resolved data for each participant.

## Data analysis

### Behavioral data

We compared differences in pain and expectation ratings for the different cue conditions by computing two repeated-measures ANOVAs with cue type (placebo vs. nocebo vs. control) as predictor and pain and expectation ratings as outcome, respectively. Partial eta-squared was used to describe effect sizes.

Furthermore, we analyzed variability within conditions indicated by the relative variability index (*Mestdagh et al., 2018*) by computing two repeated-measures ANOVA with cue type (placebo vs. nocebo vs. control) and measurement block (block 1, block 2, block 3) for the relative variability index of expectation and pain ratings, respectively.

### Electrodermal data

We compared differences in SCRs in the pain phase by conducting a repeated-measures ANOVA with the factor cue type (placebo vs. nocebo vs. control) as predictor and SCR as outcome.

### fMRI data

#### Statistical inference

For each subject, a finite impulse response model (FIR model) was set up on a time course of 18.4 s starting at the onset of the cue, divided into 11 bins, with a bin roughly covering the duration of one TR (1.679 s compared to 1.675 s). The FIR model was implemented separately for each condition. Data was also corrected for cardioballistic and respiratory artifacts by including them as regressors built with the RETROICOR algorithm of the PhysIO toolbox (*Frässle et al., 2021*; *Kasper et al., 2017*).

On the group level in a flexible factorial design, directed *t*-contrasts were set up for common effects (placebo and nocebo vs. control), exclusively masked with the *F*-contrast between placebo and nocebo (thresholded at p<0.05 uncorrected) to identify areas that showed a similar response for placebo and nocebo but different to the control condition in the anticipation phase. For the comparison between placebo and nocebo, directed *t*-contrasts were set up to identify areas that showed distinct modulation by placebo and nocebo in both the anticipation and pain phases. Analyses in the anticipation phase were performed by including the FIR regressors covering the time period from –4.275 s until –0.925 s relative to pain onset (bins 4 and 5), analyses in the pain phase by including the FIR regressors covering the time from 0.75 s until 5.8 s relative to pain onset (bins 7, 8, and 9). All analyses were corrected for multiple comparisons using FWE (p<0.05) correction.

#### ROI analyses

Additionally, ROI analyses were conducted regarding a priori hypotheses in the following areas defined by the anatomy based on the Harvard-Oxford atlas: insular cortex, thalamus, ACC, hippocampus, and amygdala. Furthermore, an ROI analysis was conducted on the DLPFC based on the clusters identified in the meta-analysis conducted by *Zunhammer et al., 2021* by applying a 15 mm-radius sphere around the two reported peak coordinates (xyz$_{MNI}$: 42, 11, 33, and xyz$_{MNI}$: –30, 13, 54) bilaterally.

#### Combined EEG-fMRI analysis

Single-trial fMRI BOLD response amplitudes were estimated based on the preprocessed MR data using GLMsingle (*Prince et al., 2022*). The hemodynamic response during the anticipation phase was estimated by fitting a boxcar function with a length of 1.679 s to the anticipation onset. The accuracy of beta estimates was improved by an adaptation of GLMdenoise for single-trial beta estimation. Furthermore, the noise was reduced by using fractional ridge regression as integrated into the GLMsingle toolbox. For each trial, we extracted the mean beta within several regions of interest centered around the significant peak voxels derived from the MR analyses of common effects of expectations during the anticipation phase. These included the left anterior insula (xyz$_{MNI}$: –28, 22,–6), left (xyz$_{MNI}$: –2, 40,–4) and right ACC (xyz$_{MNI}$: 4, 42, 12), right vmPFC (xyz$_{MNI}$: 14, 56,–14), left (xyz$_{MNI}$: –32, 18, 36) and right DLPFC (xyz$_{MNI}$: 40, 24, 36), and left thalamus (–6,–12, 4; all with 10 mm sphere). For each participant on a single-trial level, Spearman's rank correlation coefficients between beta ROI estimates and time-frequency EEG data were computed, resulting in one time-frequency-resolved correlation pattern per participant and ROI. For the group-level analysis, correlations were Fisher-z-transformed

and tested against zero using nonparametric cluster-based permutation tests as implemented in the Fieldtrip toolbox (cluster threshold: p=0.05, minimum neighbors: 2, number of randomizations: 2000). Statistics were calculated from cue onset until pain offset (–9.3 until 3.9 s relative to pain onset).

## Acknowledgements

This research is funded by the Deutsche Forschungsgemeinschaft (DFG, German Research Foundation): TRR 289 Treatment Expectation — Project Number 422744262, Project A03.

## Additional information

### Funding

| Funder | Grant reference number | Author |
|---|---|---|
| Deutsche Forschungsgemeinschaft | SFB 289 | Christoph Arne Wittkamp Maren-Isabel Wolf Michael Rose |
| Deutsche Forschungsgemeinschaft | 422744262 | Christoph Arne Wittkamp Maren-Isabel Wolf Michael Rose |
| Deutsche Forschungsgemeinschaft | Project A03 | Christoph Arne Wittkamp Maren-Isabel Wolf Michael Rose |

The funders had no role in study design, data collection and interpretation, or the decision to submit the work for publication.

### Author contributions

Christoph Arne Wittkamp, Maren-Isabel Wolf, Conceptualization, Data curation, Software, Formal analysis, Validation, Investigation, Visualization, Methodology, Writing – original draft, Project administration, Writing – review and editing; Michael Rose, Conceptualization, Resources, Software, Supervision, Funding acquisition, Validation, Methodology, Writing – original draft, Project administration, Writing – review and editing

### Author ORCIDs

Christoph Arne Wittkamp ⓘ https://orcid.org/0009-0000-8609-9053
Maren-Isabel Wolf ⓘ https://orcid.org/0000-0003-3201-0134
Michael Rose ⓘ https://orcid.org/0000-0002-9789-7066

### Ethics

The study was approved by the local ethics committee (Ethikkomission der deutschen Arztekammer Hamburg) (PV7170). Infomed consent and consent to publish was obtained.

Reviewer #1 (Public review): https://doi.org/10.7554/eLife.97793.3.sa1
Reviewer #2 (Public review): https://doi.org/10.7554/eLife.97793.3.sa2
Author response https://doi.org/10.7554/eLife.97793.3.sa3

## Additional files

### Supplementary files

• Supplementary file 1. Tables containing all fMRI contrasts and results from the combined EEG-fMRI analysis. (a) Common effects of positive and negative expectations compared to control in the anticipation phase. (b) Differential activation for expectation compared to neutral expectation in the pain phase. (c) Differential activation between placebo and nocebo in the anticipation phase. (d) Activation for positive expectations compared to negative expectations in the pain phase. (e) Combined EEG-fMRI analysis.

• MDAR checklist

## Data availability

Derived data that support the findings of this study are available at https://osf.io/3g49v/. Due to data privacy restrictions, further data is only available on request.The only data not publicly available are the unprocessed raw data, which could potentially be used to re-identify participants; this measure is in place to safeguard participant privacy. Researchers interested in accessing the raw data should contact the lead investigator Michael Rose (rose@uke.de), providing a rationale for their request. Upon review, they will be granted access to the raw data.

The following dataset was generated:

| Author(s) | Year | Dataset title | Dataset URL | Database and Identifier |
|---|---|---|---|---|
| Wittkamp CA, Wolf M, Rose M | 2024 | The neural dynamics of positive and negative expectations of pain | https://osf.io/3g49v/ | Open Science Framework, 10.17605/OSF.IO/3G49V |

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
