## [Editor Report · eLife Assessment]

Wittkamp et al. investigated the spatiotemporal dynamics of expectation of pain using an original fMRI-EEG approach. The methods are solid and the evidence for a substantially different neural representation between the anticipatory and the actual pain period is **convincing**. These **important** findings are discussed within a general framework that encompasses their research questions, hypotheses, and analysis of results. Although the choice of conditions and their influence on the results might accept different interpretations, the manuscript is strong and contributes beneficial insights to the field.

---

## [Referee Report · Reviewer #1 (Public review)]

Summary:

In this important paper the authors investigate the temporal dynamics of expectation of pain using a combined fMRI-EEG approach. More specifically, by modifying the expectations of higher or lower pain on a trial-to- trial basis they report that expectations largely share the same set of activations before the administration of the painful stimulus and that the coding of the valence of the stimulus is observed only after the nociceptive input has been presented. fMRI informed EEG analysis suggested that the temporal sequence of information processing involved the Dorsolateral prefrontal cortex (DLPFC), the anterior insula and the anterior cingulate cortex. The strength of evidence is convincing, the methods are solid, but a few alternative interpretations about the findings related to the control group, as well as a more in depth discussion on the correlations between the BOLD and EEG signals would strengthen the manuscript.

Strengths:

In line with open science principles, the article presents the data and the results in a complete and transparent fashion.

On the theoretical standpoint, the authors make a step forward in our understanding of how expectations modulate pain by introducing a combination of spatial and temporal investigation. It is becoming increasingly clear that our appraisal of the world is dynamic, guided by previous experiences and mapped on a combination of what we expect and what we get. New research methods, questions and analyses are needed to capture this evolving process.

Weaknesses:

The authors have addressed my concerns about the control condition and made some adjustments, namely acknowledging that participants cannot be "expectations" free and investigating whether scores in the control condition are simply due to a "regression to the mean".

General considerations and reflections

Inducing expectations in the desired direction is not a straightforward task, and results might depend on the exact experimental conditions and the comparison group. In this sense, the authors choice of having 3 groups of positive, negative and "neutral" expectations is to be praised. On the other hand, also control groups form their expectations, and this can constitute a confounder in every experiment using expectation manipulation, if not appropriately investigated. The authors have addressed this element in their revised submission.

In addition, although fMRI is still (probably) the best available tool we have to understand the spatial representation of cortical processing, limitations about not only the temporal but even the spatial resolution should be acknowledged. This has been done. Given the anatomical and physiological complexity of the cortical connections, as we know from the animal world, it is still well possible that sub circuits are activated also for positive and negative expectations, but cannot be observed due to the limitation of our techniques. Indeed, on an empirical/evolutionary bases, it would remain unclear why we should have a system that waits for the valence of a stimulus to show differential responses.

Also, moving in a dimension of network and graph theory, one would not expect single areas to be responsible for distinct processes, but rather that they would more integrate information in a shared way, potentially with different feedback and feedforward communications. As such, it becomes more difficult to assume the insula as a center for coding potential pain, perhaps more of a node in a system that signals potential dangers for the integrity of the body.

The rationale for the choice of their EEG band has been outlined.

---

## [Referee Report · Reviewer #2 (Public review)]

I appreciate the authors' thorough revision of the manuscript, which has significantly improved its quality. I have no additional comments or requests for further changes.

However, I remain in slight disagreement regarding the characterization of the neutral condition. My perspective is that it resembles more of a "medium" condition, making it challenging to understand what would be common to "high-medium" and "low-medium" contrasts. I suspect that the neutral condition might represent a state of high uncertainty since participants are informed that the algorithm cannot provide a prediction. From this viewpoint, the observed similarities in effects for both positive and negative expectations may actually reflect differences between certainty and uncertainty rather than the specific expectations themselves.

Nevertheless, the authors have addressed alternative interpretations of their discussion section, and I have no further requests. The paper is well-executed and demonstrates several strengths: the procedure effectively induced varying levels of expectations with clear impacts on pain ratings. Additionally, the integration of fMRI with EEG is commendable for tracking the transition from anticipatory to pain periods. Overall, the manuscript is strong and contributes valuable insights to the field.

---

## [Author Response]

The following is the authors’ response to the original reviews.

We thank the reviewers for their careful and overall positive evaluation of our work and the constructive feedback! To address the main concerns, we have:

– Clarified a major misunderstanding of our instructions: Participants were only informed that they would receive different stimuli of medium intensity and were thus not aware that the stimulation temperature remained constant

– Implemented a new analysis to evaluate how participants rated their expectation and pain levels in the control condition

– Added a paragraph in the discussion in which we argue that our paradigm is comparable to previous studies

Below, we provide responses to each of the reviewers’ comments on our manuscript.

**Reviewer #1 (Public Review):**
Summary:In this important paper, the authors investigate the temporal dynamics of expectation of pain using a combined fMRI-EEG approach. More specifically, by modifying the expectations of higher or lower pain on a trial-to-trial basis, they report that expectations largely share the same set of activations before the administration of the painful stimulus, and that the coding of the valence of the stimulus is observed only after the nociceptive input has been presented. fMRIinformed EEG analysis suggested that the temporal sequence of information processing involved the Dorsolateral prefrontal cortex (DLPFC), the anterior insula, and the anterior cingulate cortex. The strength of evidence is convincing, and the methods are solid, but a few alternative interpretations about the findings related to the control group, as well as a more in-depth discussion on the correlations between the BOLD and EEG signals would strengthen the manuscript.

Thank you for your positive evaluation! In the revised version of the manuscript, we elaborated on the control condition and the BOLD-EEG correlations in more detail.

Strengths:In line with open science principles, the article presents the data and the results in a complete and transparent fashion.From a theoretical standpoint, the authors make a step forward in our understanding of how expectations modulate pain by introducing a combination of spatial and temporal investigation. It is becoming increasingly clear that our appraisal of the world is dynamic, guided by previous experiences, and mapped on a combination of what we expect and what we get. New research methods, questions, and analyses are needed to capture these evolving processes.

Thank you very much for these positive comments!

Weaknesses:The control condition is not so straightforward. Across the manuscript it is defined as "no expectation", and in the legend of Figure 1 it is mentioned that the third state would be "no prediction". However, it is difficult to conceive that participants would not have any expectations or predictions. Indeed, in the description of the task it is mentioned that participants were instructed that they would receive stimuli during "intermediate sensitive states". The results of the pain scores and expectations might support the idea that the control condition is situated in between the placebo and nocebo conditions. However, since this control condition was not part of the initial conditioning, and participants had no reference to previous stimuli, one might expect that some ratings might have simply "regressed to the mean" for a lack of previous experience.General considerations and reflections:

Inducing expectations in the desired direction is not a straightforward task, and results might depend on the exact experimental conditions and the comparison group. In this sense, the authors' choice of having 3 groups of positive, negative, and "neutral" expectations is to be praised. On the other hand, also control groups form their expectations, and this can constitute a confounder in every experiment using expectation manipulation, if not appropriately investigated.

Thank you for raising these important concerns! Firstly, as it seems that we did not explain the experimental procedure in a clear fashion, there appeared to be a general misunderstanding regarding our instructions. We want to emphasize that we did not tell participants that the stimulus intensity would always be the same, but that pain stimuli would be different temperatures of medium intensity. Furthermore, our instruction did not necessarily imply that our algorithm detected a state of medium sensitivity, but that the algorithm would not make any prediction, e.g., due to highly fluctuating states of pain sensitivity, or no clear-cut state of high or low pain sensitivity. We changed this in the Methods (ll. 556-560, 601-606, 612-614) and Results (ll. 181-192) sections of the manuscript to clarify these important features of our procedure.

Then, we absolutely agree that participants explicitly and implicitly form expectations regarding all conditions over time, including the control condition. We carefully considered your feedback and rephrased the control condition, no longer framing it as eliciting “no expectations” but as “neutral expectations” in the revised version of the manuscript. This follows the more common phrasing in the literature and acknowledges that participants indeed build up expectations in the control condition. However, we do still think that we can meaningfully compare the placebo and nocebo condition to the control condition to investigate the neuronal underpinnings of expectation effects. Independently of whether participants build up an expectation of “medium” intensities in the control condition, which caused them to perceive stimuli in line with this expectation, or if they simply perceived the stimuli as they were (of medium intensity) with limited effects of expectations, the crucial difference to the placebo and nocebo conditions is that there was no alteration of perception due to previous experiences or verbal information and no shift of perception from the actual stimulus intensity towards any direction in the control condition. This allowed us to compare the neural basis of a modulation of pain perception in either direction to a condition in which this modulation did not take place.

**Author response image 1. sa3fig1:** Variability within conditions over time. Relative variability index for expectation (left) and pain ratings (right) per condition and measurement block.

Lastly, we want to highlight that our finding of the control condition being rated in between the placebo and nocebo condition is in line with many previous studies that included similar control conditions and advanced our understanding of pain-related expectations (Bingel et al., 2011; Colloca et al., 2010; Shih et al., 2019). We thank the reviewer for the very interesting idea to evaluate the development of ratings in the control condition in more detail and added a new analysis to the manuscript in which we compared how much intra-subject variance was within the ratings of each of the three conditions and how much this variance changed over time. For this aim, we computed the relative variability index (Mestdagh et al., 2018), a measure that quantifies intra-subject variation over multiple ratings, and compared between the three conditions and the three measurement blocks. We observed differences in variances between conditions for both expectation (*F*(2,96) = 8.14, *p* < .001) and pain ratings (*F*(2,96) = 3.41, *p* = .037). For both measures, post-hoc tests revealed that there was significantly more variance in the placebo compared to the control condition (both _p_holm < .05), but no difference between control and nocebo. The substantial and comparable variation in pain and expectation ratings in all three conditions (or at least between control and nocebo) shows that participants did not always expect and perceive the same intensity within conditions. Variance in expectation ratings decreased from the first block compared to the other two blocks (*F*(1.35,64.64) = 5.69, *p* = .012; both _p_holm < .05), which was not the case for pain ratings. Most importantly, there was no interaction effect of block and condition for neither expectation (*F*(2.65,127.06) = 0.40, *p* = .728) nor pain ratings (*F*(4,192) = 0.48, *p* = .748), which implies that expectations were similarly dynamically updated in all conditions over the course of the experiment. This speak against a “regression to the mean” in the control condition and shows that control ratings fluctuated from trial to trial. We included this analysis and a more in-depth discussion of the choice of conditions in the Result (ll. 219-232) and Discussion (ll. 452-486) sections of the revised manuscript.

In addition, although fMRI is still (probably) the best available tool we have to understand the spatial representation of cortical processing, limitations about not only the temporal but even the spatial resolution should be acknowledged. Given the anatomical and physiological complexity of the cortical connections, as we know from the animal world, it is still well possible that subcircuits are activated also for positive and negative expectations, but cannot be observed due to the limitation of our techniques. Indeed, on an empirical/evolutionary basis it would remain unclear why we should have a system that waits for the valence of a stimulus to show differential responses.

We agree that the spatial resolution of fMRI is limited and that our signal is often not able to dissociate different subcircuits. Whether on this basis differential processes occurred cannot be observed in fMRI but is indeed possible. We now include this reasoning in our Discussion (ll. 373-377):

“Importantly, the spatial resolution of fMRI is limited when it comes to discriminating whether the same pattern of activity is due to identical activation or to activation in different sub-circuits within the same area. Nonetheless, the overlap of areas is an indicator for similar processes involved in a more general preparation process.*”*

Also, moving in a dimension of network and graph theory, one would not expect single areas to be responsible for distinct processes, but rather that they would integrate information in a shared way, potentially with different feedback and feedforward communications. As such, it becomes more difficult to assume the insula is a center for coding potential pain, perhaps more of a node in a system that signals potential dangers for the integrity of the body.

We appreciate the feedback on our interpretation of our results and agree that the overall network activity most likely determines how a large part of expectations and pain are coded. We therefore adjusted the Discussion, embedding the results in an interpretation considering networks (ll. 427-430, 432-435,438-442).

The authors analyze the EEG signal between 0.5 to 128 Hz, finding significant results in the correlation between single-trial BOLD and EEG activity in the higher gamma range (see Figure 6 panel C). It would be interesting to understand the rationale for including such high frequencies in the signal, and the interpretation of the significant correlation in the high gamma range.

On a technical level, we adapted our EEG processing pipeline from Hipp et al. (2011) who similarly investigated signals up to 128 Hz. Of note, the spectral smoothing was adjusted to match 3/4 octave, meaning that the frequency resolution at 128 Hz is rather broad and does not only contain oscillations at 128 Hz sharp. Gamma oscillations in general have repeatedly been reported in relation to pain and feedforward signals reflecting noxious information (e.g. Ploner et al., 2017; Strube et al., 2021). Strube et al. (2021) reported the highest effects of pain stimulus intensity and prediction error processing at high gamma frequencies (100 and 98 Hz, respectively). These findings could also serve as basis to interpret our results in this frequency range: If anticipatory activation in the ACC is linked to high gamma oscillations, which appear to play an important role in feedforward signaling of pain intensity and prediction errors, this could indicate that later processing of intensity in this area is already pre-modulated before the stimulus actually occurs. Of note: although not significant, it looks as if the cluster extends further into pain processing on a descriptive level. We added additional explanation regarding the interpretation of the correlation in the Discussion (ll. 414425):

“The link between anticipatory activity in the ACC and EEG oscillatory activity was observed in the high gamma band, which is consistent with findings that demonstrate a connection between increased fMRI BOLD signals and a relative shift from lower to higher frequencies (Kilner et al., 2005). Gamma oscillations have been repeatedly reported in the context of pain and expectations and have been interpreted as reflecting feedforward signals of noxious information (e.g. Ploner et al., 2017; Strube et al., 2021). In combination with our findings, this might imply that high frequency oscillations may not only signal higher actual or perceived pain intensity during pain processing (Nickel et al., 2022; Ploner et al., 2017; Strube et al., 2021; Tu et al., 2016), but might also be instrumental in the transfer of directed expectations from anticipation into pain processing.”

**Reviewer #2 (Public Review):**
I think this is a very promising paper. The combination of EEG and fMRI is unique and original. However, I also have some suggestions that I think could help improve the manuscript.This manuscript reports the findings of an EEG-fMRI study (n = 50) on the effects of expectations on pain. The combination of EEG with fMRI is extremely original and well-suited to study the transition from expectation to perception. However, I think that the current treatment of the data, as well as the way that the manuscript is currently written, does not fully capitalize on the potential of this unique dataset. Several findings are presented but there is currently no clear message coming out of this manuscript.First, one positive point is that the experimental manipulation clearly worked. However, it should be noted that the instructions used are not typical of studies on placebo/nocebo. Participants were not told that the stimulations would be of higher/lower intensity. Rather, they were told that objective intensities were held constant, but that EEG recordings could be used to predict whether they would perceive the stimulus as more or less intense. I think that this is an interesting way to manipulate expectations, but there could have been more justification in the introduction for why the authors have chosen this unusual procedure.

Most importantly, we again want to emphasize again that participants were not aware that the stimulation temperature was always the same but were informed that they would receive different stimuli of medium intensity. We now clarify this in the revised Results (ll. 190-192) and Methods (ll. 612-614) sections.

While we agree that our procedure was not typical, we do not think that the manipulation is not comparable to previous studies on pain-related expectations. To our knowledge, either expectations regarding a treatment that changes pain perception (treatment expectancy) or expectations regarding stimulus intensities (stimulus expectancy) are manipulated (see Atlas & Wager, 2014). In our study, participants received a cue that induced expectations in regard to a ”treatment”, although in this case the “treatment” came from changes in their own brain activity. This is comparable to studies using TENS-devices that are supposedly changing peripheral pain transmission (Skvortsova et al., 2020). Thus, although not typical, our paradigm could be classified as targeting treatment expectancies and allowed us to examine effects on a trial-by-trial level within subjects. We added a paragraph regarding the comparability of our paradigm with previous studies in the Discussion of the revised manuscript (ll. 452-464) .

Also, the introduction mentions that little is known about potential cerebral differences between expectations of high vs. low pain expectations. I think the fear conditioning literature could be cited here. Activations in ACC, SMA, Ins, parahippocampal gyrus, PAG, etc. are often associated with upcoming threat, whereas activations vmPFC/default mode network are associated with safety.

We thank you for your suggestions to add literature on fear conditioning. We agree there is some overlap between fear conditioning and expectation effects in humans, but we also believe there are fundamental differences regarding their underlying processes and paradigms. E.g. the expectation effects are not driven by classical learning algorithms but act in a large amount as self-fulfilling prophecies (see e.g. Jepma et al., 2018). However, we now acknowledge the similarities e.g in the recruitment of the insula and the vmPFC of the modalities in our Introduction (ll. 132-136).

The fact that the authors didn't observe a clearer distinction between high and low expectations here could be related to their specific instructions that imply that the stimulus is the same and that it is the subjective perception that is expected to change. In any case, this is a relatively minor issue that is easy to address.

We apologize again for the lack of clarity in our instructions: Participants were unaware that they would receive the exact same stimulus. The clear effects of the different conditions on expectation and pain ratings also challenge the notion that participants always expected the same level of stimulation and/or perception. Additionally, if participants were indeed expecting a consistent level of intensity in all conditions, one would also assume to see the same anticipatory activation in the control condition as in the placebo and nocebo conditions, which is not the case. Thus, we respectfully disagree that the common effects might be explained by our instructions but would argue that they indeed reflect common (anticipatory) processes of positive and negative expectations.

Towards the end of the introduction, the authors present the aims of the study in mainly exploratory terms:(1) What are the differences between anticipation and perception?(2) What regions display a difference between high and low expectations (high > low or low < high) vs. an effect of expectation regardless of the direction (high and low different than neutral)?I think these are good questions, but the authors should provide more justification, or framework, for these questions. More specifically, what will they be able to conclude based on their observations?For instance (note that this is just an example to illustrate my point. I encourage the authors to come up with their own framework/predictions) :(1) Possibility #1: A certain region encodes expectations in a directed fashion (high > low) and that same region also responds to perception in the same direction (high > low). This region would therefore modulate pain by assimilating perception towards expectations.(2) Possibility # 2: different regions are involved in expectation and perception. Perhaps this could mean that certain regions influence pain processing through descending facilitation for instance...

Thank you for pointing out that our hypotheses were not crafted carefully enough. We tried to give better explanations for the possible interpretations of our hypotheses. Additionally, we interpreted our results on the background of a broader framework for placebo and nocebo effects (predictive coding) to derive possible functions of the described brain areas. We embedded this in our Introduction (ll. 74-86, 158-175) and Discussion (ll. 384-388), interpreting the anticipatory activity and the activity during pain processing in the context of expectation formation as described in Büchel et al. (2014).

Interpretation derived from our framework (ll. 384-388):

e.g.: “Following the framework of predictive coding, our results would suggest that the DPMS is the network responsible for integrating ascending signals with descending signals in the pain domain and that this process is similar for positive and negative valences during anticipation of pain but differentiates during pain processing.”

Regarding analyses, I think that examining the transition from expectations to perception is a strong angle of the manuscript given the EGG-fMRI nature of the study. However, I feel that more could have been done here. One problem is that the sequence of analyses starts by identifying an fMRI signal of interest and then attempts to find its EEG correlates. The problem is that the low temporal resolution of fMRI makes it difficult to differentiate expectation from perception, which doesn't make this analysis a good starting point in my opinion. Why not start by identifying an EEG signal that differentiates perception vs expectation, and then look for its fMRI correlates?

We appreciate your feedback on the transition from expectations to perceptions and also think that additional questions could be answered with our data set. However, based on the literature we had specific hypotheses regarding specific brain areas, and we therefore decided to start from the fMRI data with the superior spatial resolution and EEG was used to focus on the temporal dynamics within the areas important for anticipatory processes. We share the view that many different approaches in analyzing our data are possible. On the other hand, identifying relevant areas based on EEG characteristics inherits even more uncertainty due to the spatial filtering of the EEG signal. For the research question of this study a more accurate evaluation of the involved areas and the related representation was more important. We therefore decided to only implement the procedure already present in the manuscript.

Finally, I found the hypotheses on "valenced" vs. "absolute" effects a little bit more difficult to follow. This is because "neutral" is not really neutral: it falls in between low and high. If I follow correctly, participants know that the temperature is always the same. Therefore, if they are told that the machine cannot predict whether their perception is going to be low or high, then it must be because it is likely to be in between. Ratings of expectation and pain ratings confirm that. The neutral condition is not "devoid" of expectations as the authors suggest.Therefore, it would make sense to look at regions with the following pattern low > neutral > high, or vice-versa, low < neutral < high. Low & high being different than neutral is more difficult to interpret. I don't think that you can say that it reflects "absolute" expectations because neutral is also the expectation of a medium temperature. Perhaps it reflects "certainty/uncertainty" or something like that, but it is not clear that it reflects "expectations".

Thank you for your valuable feedback! We considered your concerns about the interpretation of our results and completely agree that the control condition cannot be interpreted as void of expectations (ll. 119-123). We therefore evaluated the control condition in more detail in a separate analysis (ll. 219-232) and integrated a new assessment of the conditions into the Discussion (ll. 465-486). We changed the phrasing of our control condition to “neutral expectations”, as we agree that the control condition is not void of expectations and this phrasing is more in line with other studies (e.g. Colloca et al., 2010; Freeman et al., 2015; Schmid et al., 2015). We would argue that the neutral expectations can still be meaningfully compared to positive and negative expectations because only the latter shift expectations and perception in one direction. Thus, we changed our wording throughout the manuscript to acknowledge that we indeed did not test for general effects of expectations vs. no expectations, but for effects of directed expectations. Please also see our reasoning regarding the control condition in response to Reviewer 1, in which we addressed the interpretation of the control condition. We therefore still believe that the contrasts that we calculated between conditions are valid. The proposed new contrast largely overlaps with our differential contrast low>high and vice versa already reported in the manuscript (for additional results also see Supplements).

**Recommendations for the authors:**

**Reviewer #1 (Recommendations For The Authors):**
Figure 6, panel C. The figure mentions Anterior Cingulate Cortex R, whereas the legend mentions left ACC. Please check.

Thanks for catching this, we changed the figure legend accordingly.

**Reviewer #2 (Recommendations For The Authors):**
- I don't think that activity during the rating of expectations is easily interpretable. I think I would recommend not reporting it.

The majority of participants completed the expectation rating relatively quickly (M = 2.17 s, SD = 0.35 s), which resulted in the overlap between the DLPFC EEG cluster and the expectation rating encompassing only a limited portion of the cluster (~ 1 s). We agree that this activity still is more difficult to interpret, yet we have decided to report it for reasons of completeness.

- The effects on SIIPS are interesting. I think that it is fine to present them as a "validation" of what was observed with pain ratings, but it also seems to give a direction to the analyses that the authors don't end up following. For instance, why not try other "signatures" like the NPS or signatures of pain anticipation? Also, why not try to look at EEG correlates of SIIPS? I don't think that the authors "need" to do any of that, but I just wanted to let them know that SIIPS results may stir that kind of curiosity in the readers.

While this would be indeed very interesting, these additional analyses are not directly related to our current research question. We fear that too many analyses could be confusing for the readers. Nonetheless, we are grateful for your suggestion and will implement additional brain signatures in future studies.

- The shock was calibrated to be 60%. Why not have high (70%) and low (30%) conditions at equal distances from neutral, like 80% and 40% for instance? The current design makes it hard to distinguish high from control. Perhaps the "common" effects of high + low are driven by a deactivation for low (30%)?

We appreciate your feedback! We adjusted the temperature during the test phase to counteract habituation typically happening with heat stimuli. We believe that this was a good measure as participants rated the control condition at roughly VAS 50 (M = 51.40) which was our target temperature and then would be equidistant to the VAS 70 and VAS 30 during conditioning when no habituation should have taken place yet. We further tested whether participants rated placebo and nocebo trials at equal distances from the control condition and found no existent bias for either of the conditions. To do this, we computed the individual placebo effect (control minus placebo) and nocebo effect (nocebo minus control) for each participant during the test phase and statistically compared whether they differed in terms of magnitude. There was no significant difference between placebo and nocebo effects for both expectation (placebo effect M = 14.25 vs. nocebo effect M = 17.22, *t*(49) = 1.92, *p* = .061) and pain ratings (placebo effect M = 6.52 vs. nocebo effect M = 5.40, *t*(49) = -1.11, *p* = .274). This suggests that our expectation manipulation resulted in comparable shifts in expectation and pain ratings away from the control condition for both the placebo and nocebo condition and thus hints against any bias of the conditioning temperatures. Please also note that the analysis of the common effects was masked for differences of the high and low, therefore the effects cannot be driven by one condition by itself.

- If I understand correctly, all fMRI contrasts were thresholded with FWE. This is fine, but very strict. The authors could have opted for FDR. Maybe I missed something here....

While it is true that FDR is the more liberal approach, it is not valid for spatially correlated fMRI data and is no longer available in SPM for the correction of multiple comparisons. The newly implemented topological peak based FDR correction is comparably sensitive with the FWE correction (see. Chumbley et al. BELEG). We opted for the slightly more conservative approach in our preregistration (_p_FWE < .05), therefore a change of the correction is not possible.

Altogether, I think that this is a great study. The combination of EEG and fMRI is truly unique and affords many opportunities to examine the transition from expectations to perception. The experimental manipulation of expectations seems to have worked well, and there seem to be very promising results. However, I think that more could have been done. At least, I would recommend trying to give more of a theoretical framework to help interpret the results.

We are very grateful for your positive feedback. We took your suggestion seriously and tried to implement a more general framework from the literature (see Büchel et al., 2014) to provide a better explanation for our results.

References

Atlas, L. Y., & Wager, T. D. (2014). A meta-analysis of brain mechanisms of placebo analgesia: Consistent findings and unanswered questions. *Handbook of Experimental Pharmacology*, *225*, 37–69. https://doi.org/10.1007/978-3-662-44519-8_3

Bingel, U., Wanigasekera, V., Wiech, K., Ni Mhuircheartaigh, R., Lee, M. C., Ploner, M., & Tracey, I. (2011). The effect of treatment expectation on drug efficacy: Imaging the analgesic benefit of the opioid remifentanil. *Science Translational Medicine*, *3*(70), 70ra14. https://doi.org/10.1126/scitranslmed.3001244

Büchel, C., Geuter, S., Sprenger, C., & Eippert, F. (2014). Placebo analgesia: A predictive coding perspective. *Neuron*, *81*(6), 1223–1239. https://doi.org/10.1016/j.neuron.2014.02.042

Colloca, L., Petrovic, P., Wager, T. D., Ingvar, M., & Benedetti, F. (2010). How the number of learning trials affects placebo and nocebo responses. *Pain*, *151*(2), 430–439. https://doi.org/10.1016/j.pain.2010.08.007

Freeman, S., Yu, R., Egorova, N., Chen, X., Kirsch, I., Claggett, B., Kaptchuk, T. J., Gollub, R. L., & Kong, J. (2015). Distinct neural representations of placebo and nocebo effects. *NeuroImage*, *112*, 197–207. https://doi.org/10.1016/j.neuroimage.2015.03.015

Hipp, J. F., Engel, A. K., & Siegel, M. (2011). Oscillatory synchronization in large-scale cortical networks predicts perception. *Neuron*, *69*(2), 387–396. https://doi.org/10.1016/j.neuron.2010.12.027

Jepma, M., Koban, L., van Doorn, J., Jones, M., & Wager, T. D. (2018). Behavioural and neural evidence for self-reinforcing expectancy effects on pain. *Nature Human Behaviour*, *2*(11), 838–855. https://doi.org/10.1038/s41562-018-0455-8

Kilner, J. M., Mattout, J., Henson, R., & Friston, K. J. (2005). Hemodynamic correlates of EEG: A heuristic. *NeuroImage*, *28*(1), 280–286. https://doi.org/10.1016/j.neuroimage.2005.06.008

Nickel, M. M., Tiemann, L., Hohn, V. D., May, E. S., Gil Ávila, C., Eippert, F., & Ploner, M. (2022). Temporal-spectral signaling of sensory information and expectations in the cerebral processing of pain. *Proceedings of the National Academy of Sciences of the United States of America*, *119*(1). https://doi.org/10.1073/pnas.2116616119

Ploner, M., Sorg, C., & Gross, J. (2017). Brain Rhythms of Pain. *Trends in Cognitive Sciences*, *21*(2), 100–110. https://doi.org/10.1016/j.tics.2016.12.001

Schmid, J., Bingel, U., Ritter, C., Benson, S., Schedlowski, M., Gramsch, C., Forsting, M., & Elsenbruch, S. (2015). Neural underpinnings of nocebo hyperalgesia in visceral pain: A fMRI study in healthy volunteers. *NeuroImage*, *120*, 114–122. https://doi.org/10.1016/j.neuroimage.2015.06.060

Shih, Y.‑W., Tsai, H.‑Y., Lin, F.‑S., Lin, Y.‑H., Chiang, C.‑Y., Lu, Z.‑L., & Tseng, M.‑T. (2019). Effects of Positive and Negative Expectations on Human Pain Perception Engage Separate But Interrelated and Dependently Regulated Cerebral Mechanisms. *Journal of Neuroscience*, *39*(7), 1261–1274. https://doi.org/10.1523/JNEUROSCI.2154-18.2018

Skvortsova, A., Veldhuijzen, D. S., van Middendorp, H., Colloca, L., & Evers, A. W. M. (2020). Effects of Oxytocin on Placebo and Nocebo Effects in a Pain Conditioning Paradigm: A Randomized Controlled Trial. *The Journal of Pain*, *21*(3-4), 430–439. https://doi.org/10.1016/j.jpain.2019.08.010

Strube, A., Rose, M., Fazeli, S., & Büchel, C. (2021). The temporal and spectral characteristics of expectations and prediction errors in pain and thermoception. *ELife*, *10.*
https://doi.org/10.7554/eLife.62809

Tu, Y., Zhang, Z., Tan, A., Peng, W., Hung, Y. S., Moayedi, M., Iannetti, G. D., & Hu, L. (2016). Alpha and gamma oscillation amplitudes synergistically predict the perception of forthcoming nociceptive stimuli. *Human Brain Mapping*, *37*(2), 501–514. https://doi.org/10.1002/hbm.23048